# Experimental evolution partially restores functionality of bacterial chemotaxis network with reduced number of components

Manika Kargeti[◐], Irina Kalita[◐], Sarah Hoch, Maryia Ratnikava, Wenhao Xu, Bin Ni[¤a], Ron L. Dy[¤b], Remy Colin, Victor Sourjik*

Max Planck Institute for Terrestrial Microbiology and Center for Synthetic Microbiology (SYNMIKRO), Marburg, Germany

◐ These authors equally contributed to this work.
¤a Current address: College of Resources and Environmental Science, National Academy of Agriculture Green Development, China Agricultural University, Beijing, China
¤b Current address: National Institute of Molecular Biology and Biotechnology (NIMBB), University of the Philippines Diliman, Quezon City, Philippines
* victor.sourjik@mpi-marburg.mpg.de

## Abstract

The chemotaxis signaling pathway, which enables bacteria to follow chemical gradients in their environment, is highly conserved among motile bacteria. It is assumed that *Escherichia coli* contains the minimal and non-redundant set of protein activities that are necessary for bacterial chemotaxis and nearly universally conserved among bacterial chemotaxis pathways. These include stimulus sensing, signal transduction towards the flagellar motor, and adaptation-based temporal comparisons of the environment. In this study, we show that functionality of the chemotaxis signaling pathway lacking some of its proteins can be partially regained by subjecting *E. coli* strains to experimental evolution under selection for chemotactic spreading in porous medium. While the core signaling components are indeed essential for the pathway function, the absence of auxiliary pathway proteins required for adaptation and desensitization could be compensated by specific sets of mutations affecting the other pathway components. Further characterization of the evolved strain lacking the adaptation enzyme CheR suggested that this strain utilizes an alternative mechanism of biased drift in chemical gradients, which does not rely on short-term adaptation that is normally considered a prerequisite for bacterial chemotaxis. Although the efficiency of this alternative mechanism remains below the one that can be achieved by the original memory-based chemotaxis strategy of *E. coli*, it can mediate chemotaxis not only in porous medium but also in liquid. Thus, even short-term experimental evolution of microorganisms can result in the appearance of behavioral strategies that are qualitatively different from those used by parental organisms.

**Data availability statement:** All data needed to evaluate the conclusions in the paper are present in the paper and/or the Supplementary Materials. Java codes for numerical simulations and for particle tracking are available at https://github.com/croelmiyn/ExperimentallyEvolvedStrains and https://gitlab.gwdg.de/remy.colin/particletracking2.

**Funding:** This work was supported by the Max Planck Society (to VS) and the European Research Council (grant 294761-MicRobE to VS). The funders had no role in study design, data collection and analysis, decision to publish, or preparation of the manuscript.

**Competing interests:** The authors have declared that no competing interests exist.

## Author summary

Chemotactic behavior of motile bacteria in environmental gradients is one of the most-studied models for signal transduction and information processing in biology. The chemotaxis pathway of the gut bacterium *Escherichia coli* has been assumed to possess the minimal set of activities that are necessary to mediate bacterial navigation in gradients. Here we demonstrate that the short-term experimental laboratory evolution could rewire the signaling network to restore the ability to follow chemical gradients in the absence of individual components that were previously considered essential. Subsequent characterization revealed that these bacteria evolved an alternative strategy that is markedly different from the established paradigm of bacterial chemotaxis and yet exhibits a comparable efficiency to that of the non-evolved wildtype cells. This demonstrates the surprising evolvability of bacterial signaling and behavior, with evolution over just a few hundred generations resulting in the appearance of a qualitatively different behavioral strategy.

## Introduction

Most motile bacteria can follow gradients of nutrients and other stimuli in their environment through chemotaxis, which is important for growth optimization, collective behaviors, and interactions with eukaryotic hosts [1,2]. The core of the signaling pathway mediating chemotaxis is highly conserved among prokaryotes [3,4]. *Escherichia coli* has one of the simplest chemotaxis pathways that is composed almost exclusively of evolutionary conserved proteins and became one of the most comprehensively studied signaling systems in biology [5]. The mechanism of bacterial chemotaxis has been demonstrated to rely on temporal comparisons of the perceived alterations in environmental conditions by swimming bacteria, where the chemotaxis signaling system determines whether the bacterium should persist in its current direction of movement or reorient itself [6]. This strategy requires two modules: one for rapid environmental sensing and signal transduction and another for slower adaptation that enables short-termed temporal comparisons of environmental conditions [5,7].

The sensory module of *E. coli* chemotaxis pathway (Fig 1A) comprises transmembrane chemoreceptors, also known as methyl-accepting chemotaxis proteins, that control the autophosphorylation activity of the receptor-associated kinase CheA with the assistance of the scaffolding protein CheW [8]. The sensory module's output is transmitted to the flagellar motor through the CheA-dependent phosphorylation of the response regulator CheY. The phosphorylation of CheY and its binding to flagellar motors increase when the bacterium travels in an unfavorable direction. This induces a switch in the motor rotation from the default counterclockwise (CCW) to clockwise (CW) direction, resulting in the flagellar bundle falling apart and the bacterium tumbling and reorienting. When swimming in a favorable direction, such as traveling up

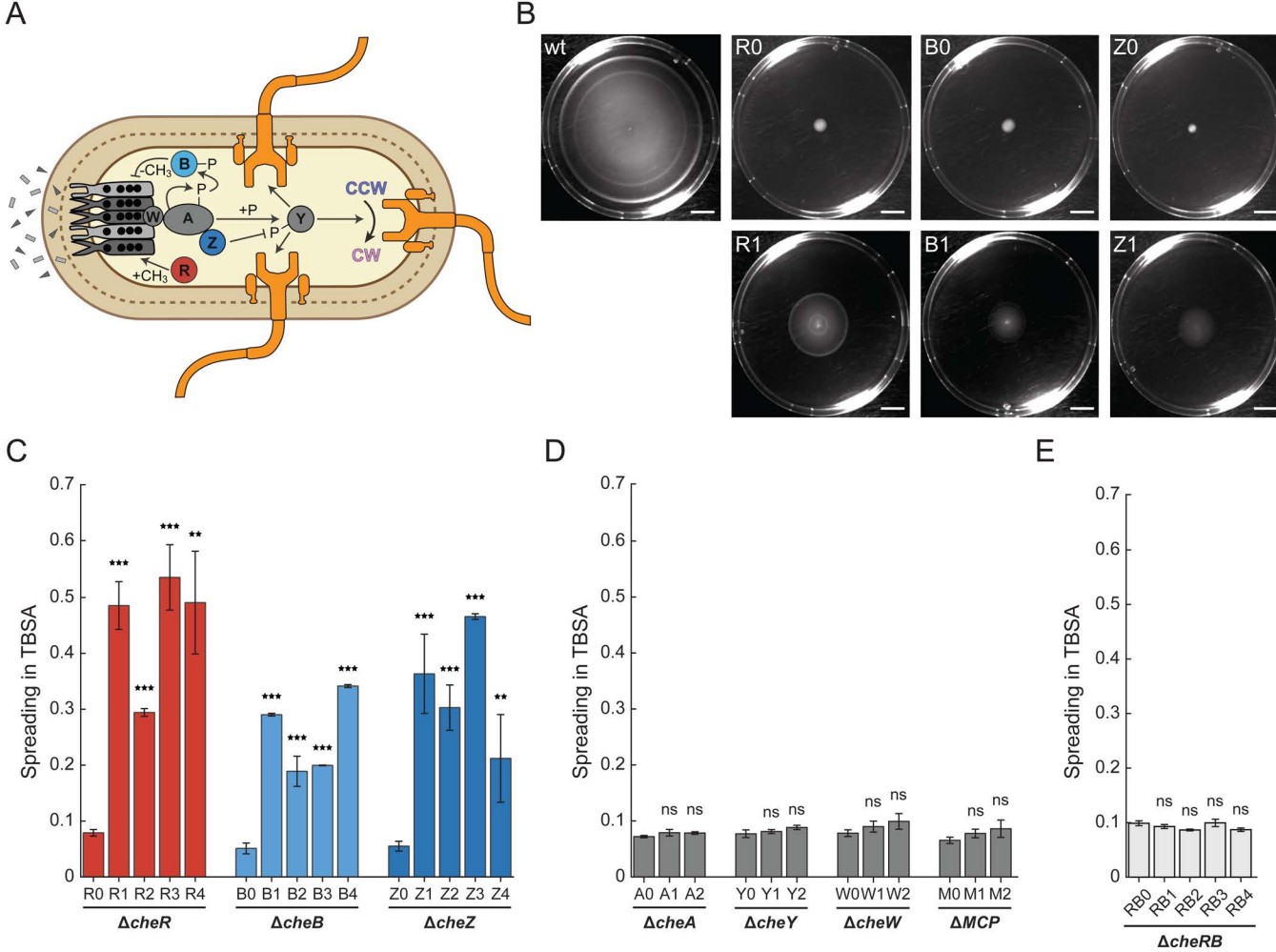

**Fig 1. Experimental evolution partly restores spreading of chemotaxis mutants in soft agar.** (A) Schematic representation of the chemotaxis signaling pathway of *Escherichia coli*. The pathway includes transmembrane chemoreceptors (two major chemoreceptors, Tar and Tsr, are shown) that form sensory complexes together with a kinase CheA (A) and an adaptor CheW (W). CheY (Y) phosphorylation by CheA mediates signal transduction to flagellar motor, inducing a switch from the default counterclockwise (CCW) to clockwise (CW) rotation. The adaptation module, including CheR (R; red) and CheB (B; light blue), regulates chemoreceptor activity through methylation and demethylation of chemoreceptors on four specific glutamates (black circles). The phosphatase CheZ (Z; dark blue) is responsible for rapid CheY dephosphorylation. See text for more details. (B) Spreading of the wildtype (wt) *E. coli* RP437 strain and of its derivatives, Δ*cheR* (R), Δ*cheB* (B) and Δ*cheZ* (Z), where the corresponding individual chemotaxis gene was deleted, in TB soft agar (TBSA) after incubation for ~16 h at 30°C. The upper row shows spreading of the non-evolved wildtype and parental deletion strains (denoted by "0"). The lower row shows one line evolved for spreading in TBSA for 30 days (line 1; denoted by "1") for each strain. Scale bars are 2 cm. (C-E) Size of the spreading rings for Δ*cheR* (R), Δ*cheB* (B) and Δ*cheZ* (Z) strains (C); for Δ*cheA* (A), Δ*cheY* (Y), Δ*cheW* (W) and receptor-less Δ*mcp* (M) strains (D); and for Δ(*cheR cheB*) (RB) strain (E) normalized to that of the wildtype *E. coli* RP437. Colors in (C) correspond to protein colors in (A). Spreading was measured for three independent replicates after incubation for ~16 h in TBSA. Several independent lines of evolution (indicated by numbers) are shown for each strain. Error bars indicate standard errors of the mean. *P* values were calculated for comparisons between spreading of evolved and respective non-evolved strains for each deletion, using two-tailed Student t-test (ns, not significant; *, *P*<0.05; **, *P*<0.01; ***, *P*<0.001).

the gradient of attractant, the binding of attractant to receptors inhibits CheA autophosphorylation, which reduces CheY phosphorylation and, in turn, favors CCW rotation and smooth swimming. This core of the sensory and signaling module is conserved in all bacterial chemotaxis systems and is evolutionary related to the broader class of bacterial two-component pathways [9]. In addition, the chemotaxis signaling module of *E. coli* and closely related proteobacteria

includes the phosphatase CheZ that is responsible for the rapid dephosphorylation of CheY, whereas other chemotaxis systems contain alternative phosphatases.

The adaptation module comprises two enzymes, the methyltransferase CheR and the methylesterase CheB, which respectively methylate or demethylate four (or five) specific glutamates on chemoreceptors. Methylated glutamates promote a high activity state of chemoreceptors. Notably, the receptors are first expressed in the intermediate activity state, with two of the four methylation sites being encoded as glutamines, which function similarly to methylated glutamates and are deamidated by CheB to glutamates. This adaptation module is unique among bacterial two-component signaling pathways but it is nearly universally present in studied chemotaxis systems, with the notable exception of gastric species of *Helicobacter* [10]. Enzymatic activity of the methylation enzymes depends on the receptor activity state, and the resulting negative feedback ensures that the steady-state activity of the pathway can adapt to intermediate level even in the presence of persistent stimulation. Additionally, changes in methylation occur with a delay following receptor stimulation, creating a short-term memory that swimming bacteria use for temporal comparisons of environmental conditions. Both functions of the adaptation module are assumed to be essential for the bacterial chemotaxis strategy.

Since chemotaxis protein activities in *E. coli* are non-redundant and, except for CheZ, nearly universally conserved in bacterial chemotaxis pathways, it is typically assumed that all of them are necessary for efficient gradient navigation by swimming bacteria [5]. Indeed, already early experimental studies have shown that *E. coli* chemotaxis requires all cytoplasmic chemotaxis proteins and at least one major chemoreceptor, Tar or Tsr [11]. Strains with deletions in *cheW*, *cheA*, *cheY*, or all receptor genes do not phosphorylate CheY, resulting in continuous running without reorientations. Conversely, *cheZ*-deficient cells have an excess of phosphorylated CheY (CheY-P) and tumble most of the time. Deletions in *cheR* or *cheB* genes also result in, respectively, very low or high levels of pathway activity. In addition to their effects on the cell tumbling bias, lack of these adaptation enzymes disables temporal comparisons, making bacteria unable to efficiently navigate chemical gradients in liquid [12]. All chemotaxis mutants also have a deficiency in spreading on soft agar plates [13], which is a commonly used assay for motility and chemotaxis that relies on the spreading of motile bacteria through agar pores following self-generated gradients of consumed chemoattractant nutrients [14].

Despite their apparent inability to perform chemotaxis, early studies indicated that *E. coli* strains lacking both CheR and CheB activities may exhibit some degree of tactic behavior [13,15]. This was further supported by the emergence of spontaneous (pseudo)revertants of the *cheR* deletion strain that could spread on soft-agar plates, with compensatory mutations mapping to either *cheB* [16] or *tsr* [17] genes. However, the compensatory mechanisms underlying this phenomenon remained unclear [17], and a subsequent study concluded that the *cheR cheB* mutants or the *cheR* revertants may rather spread in soft agar in a chemotaxis-independent fashion due to their intermediate tumbling bias, which enables slow and non-directional movement through the agar pores [14]. Such pseudotactic mutants, which carry mutations in genes encoding flagellar hook or motor proteins, have also been isolated in chemotaxis-deficient strains of other bacteria [18].

To test the essentiality of *E. coli* pathway proteins, here we subjected a set of *E. coli* deletion strains to experimental evolution under selection for chemotaxis-driven spreading in soft agar over several hundred generations. Experimental evolution, also known as adaptive laboratory evolution, is a powerful approach for investigating how individual proteins and gene regulatory networks adapt under defined selection pressure [19,20]. Recently, this approach has been used to examine the evolvability of genetic regulation under selection for motility [21] and the underlying cost-benefit tradeoffs between motility and growth [22–24]. We show that the absence of auxiliary chemotaxis proteins can be reproducibly compensated for by short-term adaptive evolution, albeit to various extent, while the core signaling functions remain essential. Importantly, the evolved strains not only regained the ability to spread in soft agar but also demonstrated biased drift in chemical gradients in soft agar and in liquid, indicating their capability to perform true chemotaxis. Analysis of the evolved ΔcheR cells revealed that their restored chemotaxis does not require short-term adaptation at the receptor level, indicating emergence of an alternative behavioral strategy.

## Results

### Experimental evolution can compensate for defects caused by the deletions of chemotaxis genes

Experimental evolution of *E. coli* mutant strains was performed under selection for increased spreading on tryptone broth soft-agar (TBSA) plates for 30 cycles of up to 16 hours each (S1A Fig), and with up to four independently evolved lines. All these strains were derived from *E. coli* strain RP437 that is commonly used as the wildtype for studies of chemotaxis [25]. We relied on the natural mutation rate of *E. coli* cells, which is known to be sufficiently high to enable evolutionary tuning of motility under strong selection for chemotaxis in TBSA [22].

Consistent with numerous previous studies, none of these strains deleted for individual chemotaxis genes initially exhibited spreading in soft agar (Figs 1B-D, and S1B). However, the spreading of the evolved Δ*cheR*, Δ*cheB*, and Δ*cheZ* strains significantly improved compared to the original deletion strains in all evolved lines (Fig 1B). These three strains lack the auxiliary components of the signaling pathway, either the adaptation enzymes or the phosphatase (Fig 1A). The extent of the improvement varied between gene deletions and lines, with the largest improvement being observed for Δ*cheR* lines where spreading reached ~50% of the non-evolved wildtype (wt) (Fig 1C). Notably, a previous study has shown that the spreading of the RP437 wildtype could itself be enhanced up to 50% by the experimental evolution under similar conditions [22], meaning that the evolved Δ*cheR* lines in our experiments could achieve ~30% spreading of the evolved wildtype.

In contrast, the absence of universally conserved core components of the chemotaxis pathway that are required for sensing and signal transduction to the motor, including CheA, CheY, CheW, or all chemotaxis receptors (MCPs), could not be compensated for by such short-term evolution (Figs 1D and S1B). Furthermore, no improvement in the spreading of a double Δ(*cheR cheB*) deletion strain under selection was observed (Fig 1E). This was surprising, since the Δ(*cheR cheB*) strain prior to evolution was spreading slightly better than the individual *cheR* or *cheB* deletion strains, likely because of its intermediate tumbling bias [14]. As the Δ*cheR* strain showed the largest enhancement of spreading, we evolved four additional Δ*cheR* lines (S2 Fig).

### Evolved strains exhibit compensatory changes in motility

We next investigated changes in the motility phenotypes of the evolved *E. coli* strains, by tracking cell swimming in liquid. Consistent with the known importance of intermediate tumbling frequency for spreading in soft agar [14,26], the evolved strains exhibited compensation for the defects in tumbling that were present in the original deletion strains. In nearly all cases, the fraction of time that cells spent tumbling became more similar to the wildtype (Fig 2A), with all evolved Δ*cheR* lines showing increased tumbling and all evolved Δ*cheZ* lines showing decreased tumbling compared to the respective non-evolved deletion strains. Cell tumbling in liquid correlated well with the increased spreading in TBSA (Fig 2B), consistent with a recent report showing that the tumbling bias of wildtype *E. coli* cells is approximately optimal for spreading in soft agar [26]. However, the evolutionary adjustment of tumbling was clearly not the sole determinant of the improved spreading, as strains with similar tumbling could spread very differently. This is exemplified by Δ*cheB* lines, where spreading of several lines has improved without changes in tumbling.

Furthermore, nearly all of the evolved strains exhibited an increased swimming velocity (Fig 2C). Similar increase in velocity was previously observed during the evolution of wildtype cells for the spreading in TBSA, and it was shown to be the consequence of the elevated expression of the flagellin gene *fliC* and other flagellar genes [22]. Therefore, we measured activity of the transcriptional reporter of *fliC* promoter (P*fliC*) [22]. This activity was indeed significantly higher in most of the evolved strains compared to their parental strains (Fig 2D), consistent with our hypothesis that the increased swimming velocity of the evolved strains may be primarily due to the changes in flagellar gene expression. Cell swimming velocity was previously shown to initially increase with the flagellar regulon activity above that in the wildtype RP437 but subsequently saturate at approximately twofold higher expression levels [22,27], which could be observed in our data as well (Fig 2D *Inset*).

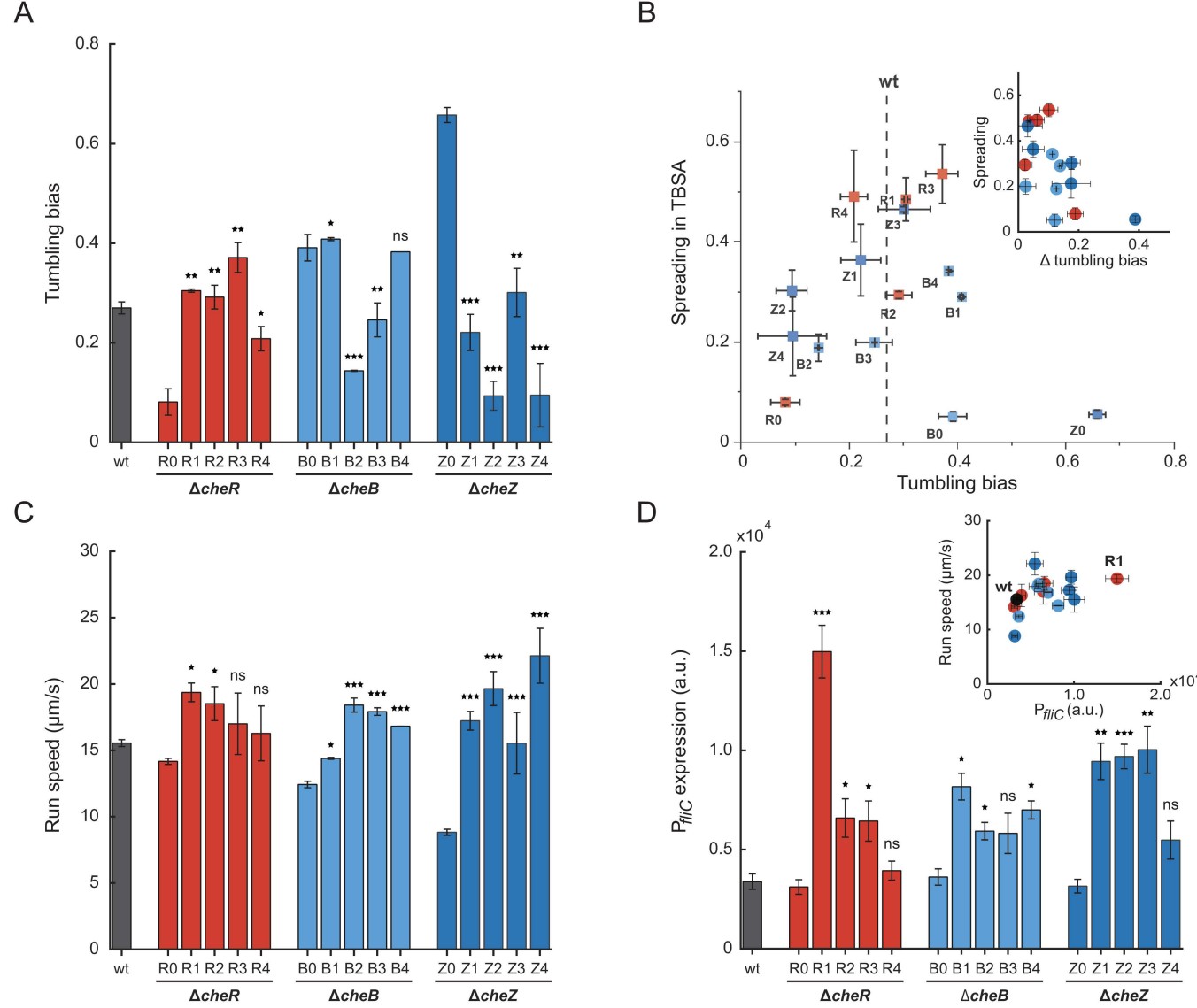

**Fig 2. Acquired changes in motility phenotypes and flagellar gene expression in the evolved lines.** (A) Tumbling bias, defined as the fraction of time spent tumbling, for the wildtype, parental non-evolved strains (R0, B0 and Z0), and evolved strains (R1-R4, B1-B4, Z1-Z4), measured in three independent replicates. (B) Size of the spreading rings in TBSA, normalized to that of the wildtype strain (data from Fig 1C), plotted as a function of tumbling bias for individual strains. The dotted line indicates the tumbling bias for the wildtype. *Inset:* Spreading in TBSA as a function of the absolute deviation of the tumbling bias from that of the wildtype strain. The relationship between the two variables was evaluated using Spearman's rank correlation, yielding a coefficient of –0.51. (C) Run speed between two consecutive tumbles, measured for all strains in three independent replicates. Motility phenotypes were assessed using cell tracking (see Materials and Methods). (D) Activity of transcriptional *fliC* promoter (P*fliC*) reporter, measured as fluorescence of green fluorescent protein (GFP) using flow cytometry in three independent replicates. *Inset:* Run speed (data from C) plotted as a function of the *fliC* promoter activity. Colors correspond to protein colors in Fig 1A. Significance analysis was done in comparison to the respective non-evolved deletion strains. Error bars indicate standard errors of the mean. *P* values were calculated using two-tailed Student t-test (ns, not significant; *, $P<0.05$; **, $P<0.01$; ***, $P<0.001$).

Since spreading in soft agar requires not only motility and chemotaxis but also cell growth, we further quantified growth of the evolved strains. We observed that their growth rate generally decreased compared to the parental strains (S3 Fig). This observation could be at least partly explained by the known negative tradeoff between expression of flagellar genes

and *E. coli* growth [22–24,27–29], and previous experimental evolution of the wildtype in TBSA indeed led to the elevated motility at the cost of slower growth [22]. Nevertheless, there was no simple correspondence between *fliC* promoter activity and growth of individual evolved strains, suggesting that other factors besides flagellar gene expression contribute to the reduction of their growth rate.

## Deletion-specific sets of compensatory mutations are observed in the evolved strains

Whole-genome sequencing revealed multiple mutations in the evolved Δ*cheR*, Δ*cheB* and Δ*cheZ* lines (S1, S2 and S3 Tables and S1 Data). These mutations exhibited clear gene-deletion specific patterns and primarily affected chemotaxis or motility genes. The most prominent set of mutations in the evolved Δ*cheR* lines mapped to the *tsr* gene that encodes the serine chemoreceptor (Fig 3A and S1 Table). Mutations in *tsr* were observed in seven out of eight lines. One line (R7) carried instead a mutation in *tar*, which encodes the aspartate receptor (Fig 3B), and two lines (R3 and R8) carried mutations in both *tsr* and *tar* (S1 Table). The corresponding amino acid substitutions affected different receptor domains, including the ligand-binding domain, transmembrane helix, HAMP domain, methylation region, and in the signaling domain, which is similar to substitutions observed in the previous report [17]. We hypothesized that these mutations may promote the active state of Tsr or Tar. This could increase the pathway activity in the evolved Δ*cheR* lines relative to the parental Δ*cheR* strain, thus contributing to the restoration of their tumbling (Fig 2A). M249 and L263 residues of Tsr were previously shown to be critical for the helical packing of the HAMP bundle and for signal transduction [30], and T305 is located adjacent to one of the methylation sites that control receptor activity [31]. Amino acids substitutions at these residues are thus likely to affect the activity state of Tsr, and other substitutions in Tsr and in Tar might be similarly activatory. Notably, mutations in *tsr* may have been preferentially selected over those in *tar* because the Tsr ligand serine is consumed faster than other amino acids [32], and the gradient of serine is thus the first gradient followed by spreading chemotactic *E. coli* cells on TBSA plates [33].

Most of the evolved Δ*cheR* lines also had mutations in the *cheB* gene (Fig 3C and S1 Table). CheB consists of a regulatory CheY-like receiver domain, which is phosphorylated by CheA, and the enzymatic methylesterase domain [34]. Mutations were found in both domains, with an apparent clustering near the catalytic pocket, the phosphorylation pocket, and the regulatory interface [35]. These mutations may lower the methylesterase activity of CheB, as previously suggested [16], which could potentially compensate for the absence of methyltransferase activity. Nevertheless, no amber or frameshift mutations were detected, indicating that certain level of CheB activity is necessary for the restored spreading of the evolved Δ*cheR* lines (see below). Finally, two of the Δ*cheR* lines had mutations in *cheZ* (Fig 3D). Q204L affects the C-terminal CheY-binding peptide [36] and Q64L is close to the catalytic site of CheZ [37]. Substitutions at both sites were reported to lower the phosphatase activity of CheZ [38], suggesting that they can partly offset the low kinase activity in the Δ*cheR* strain and thus increase the tumbling bias in the evolved lines.

Mutations in the receptor genes *tsr*, *tar* and *tap* were also found in all Δ*cheB* lines (Fig 3A and 3B, and S2 Table). Since the lack of CheB causes excessive receptor methylation and therefore hyperactivity, we would expect these mutations to counter this lack by favoring the low activity state of receptors. Additionally, one amino acid substitution affected the dimerization domain of CheA, and a short ALGD amino acid sequence was inserted in CheW (Fig 3E and S2 Table). These mutations may similarly affect the activity or stability of the ternary receptor-CheA-CheW complex, offsetting the hyperactive receptor phenotype caused by the Δ*cheB* deletion.

The evolved Δ*cheZ* lines have similarly compensated the hyperactive pathway phenotype but with a different set of mutations (S3 Table). In all lines, Tar translation was interrupted either by a stop codon mutation within the ligand-binding domain or by a frameshift in the transmembrane helix of the receptor (Fig 3B). These mutations likely lower the overall pathway activity by reducing the overall number of receptor proteins. Additionally, the disruption of *tar* results in more uniform composition of the chemosensory complexes that primarily consist of the remaining major receptor Tsr, which may increase sensitivity of the chemotactic response to the gradient of Tsr ligand serine [39] that is the first attractant gradient

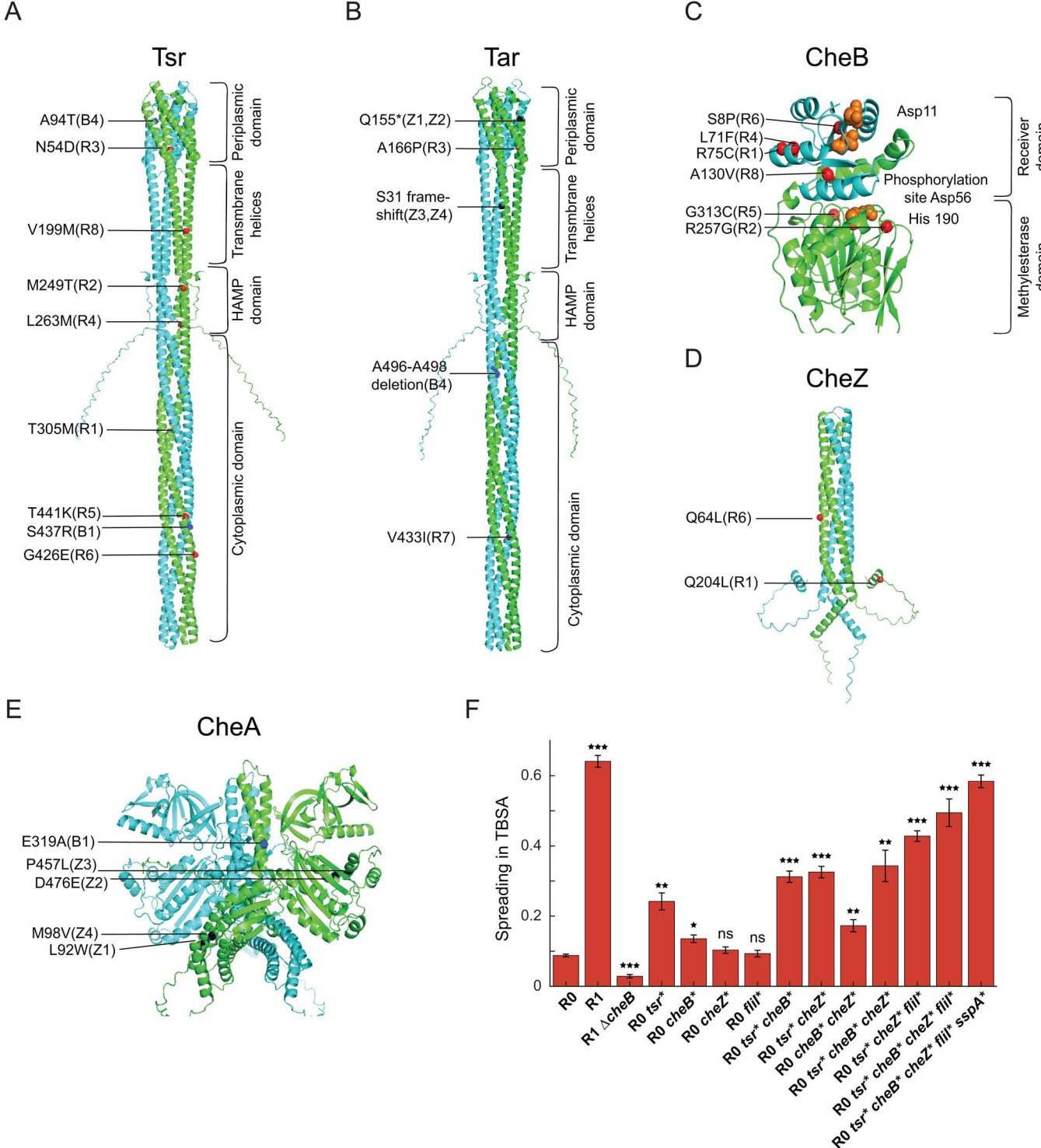

**Fig 3. Evolutionary selected amino acid substitutions in chemotaxis proteins.** (A-E) Substitutions identified in the evolved R strains (red), B strains (dark blue) and Z strains (black) in (A) Tsr, (B) Tar, (C) CheB, (D) CheZ, and (E) CheA, mapped on the respective protein structure. For chemoreceptors, functional domains are labeled. For CheB, the receiver and methylesterase domains are labeled and residues in the phosphorylation and catalytic pockets are marked in orange. (F) Size of the spreading rings in TBSA of the parental Δ*cheR* (R0) strain carrying either individual mutations that were identified in the R1 strain (indicated by asterisks; see S1 Table) or their combinations, compared to spreading of R0 and R1 strains and of R1 strain

carrying deletion of *cheB*. Values were measured in three independent replicates and normalized to the spreading ring size of the wildtype strain. Error bars indicate standard errors of the mean. Significance analysis was done in comparison to R0. *P* values were calculated using two-tailed Student t-test (ns, not significant; *, *P*<0.05; **, *P*<0.01; ***, *P*<0.001).

encountered by the spreading colony (see above). Each of these mutations was observed in two of the evolution lines, possibly because they were already present at low frequency before evolution, due to standing genetic variation within population. All Δ*cheZ* lines also had mutations in *cheA*, either in the P1 (phosphorylation) domain or in the P4 (kinase) domain (Fig 3E). These mutations may reduce the activity of CheA and consequently lower the phosphorylation of CheY, thus compensating for the absence of CheZ.

In addition to these mutations in the chemotaxis genes, almost all Δ*cheR* lines and one Δ*cheB* line had mutations in the genes that encode the export apparatus (*fliI*) and the basal body (*fliF*, *fliM, fliN*, and *fliG*) of the flagellar motor (S1 and S2 Tables). Similar mutations have been previously shown to upregulate the expression of flagellar genes by enhancing secretion of the negative regulator (anti-sigma factor) FlgM [22], which is consistent with the elevated activity of P*fliC* reporter in these lines (Fig 2D). However, amino acid substitutions in FliM, FliN and FliG might also affect cell tumbling. Some evolved lines had no mutations in the export apparatus or in the basal body genes yet showed increased flagellar gene expression, possibly due to mutations or insertions in other genes such as *clpX*, *sspA* or *rpoD*, which are known to affect the expression of the flagellar regulon [22,40] (S1, S2 and S3 Tables). Finally, the mutations in *atp* genes encoding the PMF-dependent ATP synthase may increase motility by elevating the proton motive force.

## Chemotactic spreading is restored by cumulative effects of multiple mutations

We further investigated the order in which mutations were detected over the course of evolution for several Δ*cheR* lines (S4A, S4B, S4C and S4D Fig), which revealed that the mutations in *tsr* were selected first, followed by mutations in other chemotaxis and/or flagellar genes. One of the best-spreading lines, R1, was subsequently selected to evaluate the phenotypic impacts of individual mutations and their potential epistatic interactions. R1 carries mutations in the chemotaxis genes *tsr*, *cheB* and *cheZ* (Fig 3A, 3C and 3D), and a mutation in the flagellar export gene *fliI*. When these mutations were introduced individually into the Δ*cheR* strain, mutations in *tsr* or *cheB* significantly increased spreading on TBSA plates (Fig 3F). Indeed, these two genes are most commonly affected in the evolved Δ*cheR* lines. Spreading was further increased by combinations of multiple chemotaxis and *fliI* mutations. Interestingly, the observed order of selection for the individual mutations during evolution of the R1 line (S4A Fig) is apparently consistent with the path of the largest stepwise increase in spreading due to addition of each subsequent mutation, *tsr* < *tsr cheZ* < *tsr cheZ fliI* < *tsr cheB cheZ fliI*. Spreading of the Δ*cheR tsr cheB cheZ fliI* strain largely recapitulated that of the R1 line, with the residual difference being likely due to additional mutations present in the R1 line. Indeed, inactivation of *sspA* – which was interrupted in R1 – was previously shown to increase flagellar gene expression [22], and the introduction of this mutation led to further increase in spreading (Fig 3F). A gradual increase in spreading was similarly observed when *cheB* and *tsr* mutations from R4 and R5 lines were introduced individually into the Δ*cheR* strain (S4E Fig). Similar to the R1-specific mutations, the effect of *tsr* mutations on spreading was stronger than that of *cheB* mutations.

We tested the effects of R1-specific mutations on the chemotactic spreading when introduced individually in the wildtype cells (S4F Fig). Mutation in *fliI* led to enhanced spreading, consistent with the previous report where similar mutations were shown to increase swimming velocity [22], while all other mutations resulted in no or only modest changes in spreading in the wildtype background. This indicates that all of the mutated genes remain functional, including mutated *cheB* and *cheZ* genes that are expected to produce proteins with reduced enzymatic activities to compensate for the lack of CheR. To directly test whether the function of CheB was required for the spreading of the R1 line, we introduced the *cheB*

deletion in the evolved strain. The R1 strain lacking *cheB* showed even less spreading in soft agar than the Δ*cheR* strain (Fig 3F), confirming that the (reduced) activity of CheB is necessary for the re-evolved spreading of Δ*cheR* strains.

To verify the impact of mutations in *cheB* and in *tsr* genes on the functionality of their products, we characterized the pathway response in the wildtype cells carrying the R1-specific mutations in these genes (S5 Fig). This was done using the previously established assay based on Förster (fluorescence) resonance energy transfer (FRET), which monitors the phosphorylation-dependent interaction between CheY fused to a yellow fluorescent protein (CheY-YFP) and its phosphatase CheZ fused to a cyan fluorescent protein (CheZ-CFP) [41,42]. High pathway activity leads to increased phosphorylation of CheY-YFP, which results in increased complex formation with CheZ-CFP and higher energy transfer from the excited CFP donor to the YFP acceptor fluorophore, reflected by the increased ratio between the YFP and CFP fluorescence emission. Consistent with the previous work [41], inhibition of the pathway activity in the wildtype cells by stimulation with attractant (MeAsp) leads to the adaptive pathway response, characterized by the initial decrease in FRET and subsequent recovery of the pathway activity upon prolonged stimulation (S5A Fig). This recovery is mediated by the CheR-dependent receptor methylation [41]. It is followed by a characteristic overshoot of the pathway activity upon the removal of attractant, also followed by the recovery of activity, this time due to the receptor demethylation mediated by CheB. Although the R1-specific mutation in *cheB* (R75C) introduced in the wildtype cells did not reduce their spreading in soft agar (S4F Fig), it had a clear impact on the pathway response (S5B Fig). The strain carrying this mutation showed larger amplitude of response to the addition of saturating concentration of attractant and weaker amplitude of response to its removal, suggesting that it has higher steady-state pathway activity. Moreover, this strain showed markedly slower adaptation to the removal of attractant, mediated by CheB. These observations are consistent with our expectation that this mutation reduces the activity of CheB. Similar elevation of the adapted pathway activity was also observed upon the introduction of the R1-specific mutation in *tsr* (T305M) (S5C, S5D, S5E, S5F and S5G Fig). Moreover, this latter mutation affected the dose dependence of the pathway response to the Tsr ligand serine, shifting it to higher concentrations and making it apparently steeper compared to the wildtype response (S5C, S5D and S5G Fig). In contrast, it had little effect on the dose dependence of the response to MeAsp, the ligand of Tar (S5E, S5F and S5G Fig).

## Evolved strains exhibit directional spreading in chemical gradients

Although previous studies concluded that spreading in TBSA does not necessarily require chemotaxis but only an intermediate cell tumbling frequency, such pseudotaxis is much slower than the chemotaxis-driven spreading [14,43,44]. In contrast to that, spreading of some of the evolved strains, in particular of several Δ*cheR* lines, was comparable to that of the wildtype cells, including a characteristic ring at the edge of the spreading colony that normally indicates chemotactic behavior [43,44]. We thus tested the ability of these strains to follow chemical gradients, first using the M9 minimal medium soft agar (M9SA) gradient plates with the pre-established gradient of chemoeffector [45,46]. Three out of four Δ*cheR* lines showed efficient biased spreading up the gradient of serine in M9SA plates that was similar to that of the wildtype strain, again forming a characteristic sharp ring at the edge of the colony spreading up the gradient of serine (S6A and S6D Fig). Biased spreading was also observed for two out of four Δ*cheZ* lines (S6C and S6D Fig), although it was less pronounced than for the Δ*cheR* lines. The other Δ*cheR* and Δ*cheZ* lines, as well as all Δ*cheB* lines (S6A, S6B and S6C Fig) showed little spreading or growth on M9SA plates, so their bias could not be determined.

Although the observed spreading bias in serine gradient is indicative of chemotaxis, its interpretation is complicated by the fact that serine is metabolized. We thus tested spreading of several Δ*cheR* lines using M9SA plates with a gradient of α-methyl-D, L-aspartate (MeAsp), a non-metabolizable analogue of aspartate. Indeed, a reproducible and significant biased movement up the gradient of MeAsp could be observed for the R1, R4, and R5 lines, but not for R0 (ancestor) (Fig 4A and 4B). Notably, this spreading bias of these Δ*cheR* lines was weaker than in the serine gradient and no sharp ring at the spreading edge was visible. This difference might be due to the lower steepness of the MeAsp gradient in soft agar, as it is not metabolized by the spreading colony, or to different impacts of the selected mutations in *tsr* on the pathway

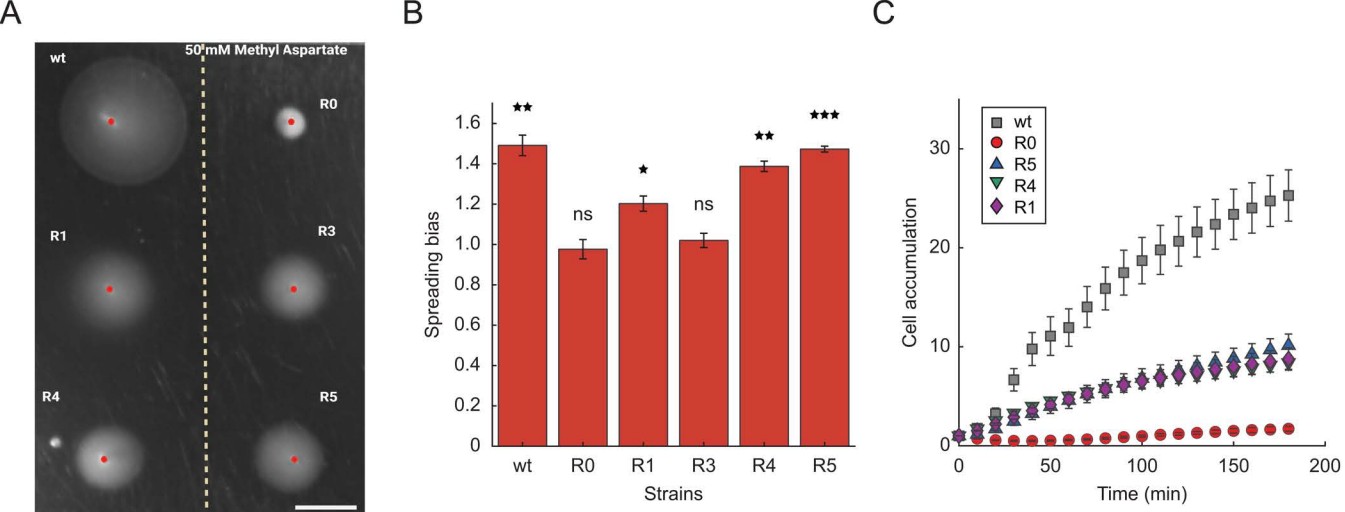

**Fig 4. Biased movement of the evolved Δ*cheR* strains towards sources of a non-metabolizable chemoattractant.** (A) Indicated strains were tested for biased spreading on the M9 minimal medium soft-agar (M9SA) plates with a pre-established gradient of α-methyl-D, L-aspartate (MeAsp), a non-metabolizable analog of aspartate, with the concentration of 50 mM at the source (dashed line). Scale bar is 1 cm. (B) Spreading bias was quantified as the ratio between the distances from the inoculation point of the expanding colony to its edges facing up and down the gradient, and measured in three independent replicates. Error bars indicate standard errors of the mean. Significance analysis was done in comparison to 1 (no bias). *P* values were calculated using one-tailed Student t-test (ns, not significant; *, $P < 0.05$; **, $P < 0.01$; ***, $P < 0.001$). (C) Chemotactic cell accumulation of the fluorescently labelled wildtype, Δ*cheR* and evolved Δ*cheR* strains towards 50 mM MeAsp at the source in the microfluidic device shown in S7A Fig. Examples of accumulation for the wildtype, R0, R1 strains are shown in S7B Fig. Total fluorescence intensity was quantified in the observation area (depicted in the dashed rectangle in S7A Fig) at indicated time points and normalized to the fluorescence at the initial time point. Error bars indicate standard errors of the mean.

response to serine and to MeAsp, as observed for the R1-specific mutation (S5 Fig). No spreading was observed for the R3 line that carries an amino acid substitution (A166P) in the ligand binding domain of Tar (Fig 3B and S1 Table), which might render it unable to sense MeAsp. Similarly, all Δ*cheZ* lines possess only a truncated version of Tar (Fig 3B and S3 Table), and therefore cannot sense MeAsp.

To next test the ability of evolved strains to perform chemotaxis in liquid, we used a microfluidic accumulation assay where an attractant gradient is generated in the liquid medium within the test channel [45–47] (S7A Fig). The wildtype strain accumulated rapidly toward the source of MeAsp in this assay, in contrast to the fully motile but non-chemotactic R0 strain, whereas an intermediate but clear accumulation was observed for the evolved Δ*cheR* lines (Figs 4C and S7B). These results demonstrate that the evolved cells lacking this normally essential pathway component show biased spreading in attractant gradients in both, soft agar and liquid, suggesting that they regained the ability to perform chemotaxis, albeit less efficiently than the wildtype cells.

### Chemotactic drift of the evolved *cheR* strain relies on greater extension of runs up the gradient without short-term adaptation

To gain an insight into the mechanism of this re-evolved chemotactic behavior, we analyzed the movement of individual swimming cells of the R1 line in a microfluidic chemotactic chamber with or without a linear gradient of MeAsp (S8A and S8B Fig). Expectedly, the R1 cells, as well as the wildtype and R0 cells, showed no biased motion, and thus no chemotactic drift, in the absence of a gradient (Fig 5A). The R0 cells also showed no bias in the gradient of MeAsp, confirming again that these cells are not capable of performing chemotaxis. In contrast, cells of the R1 line showed a clearly pronounced chemotactic drift, particularly in the gradient with 1 mM MeAsp at the source (Fig 5A). This drift could be largely

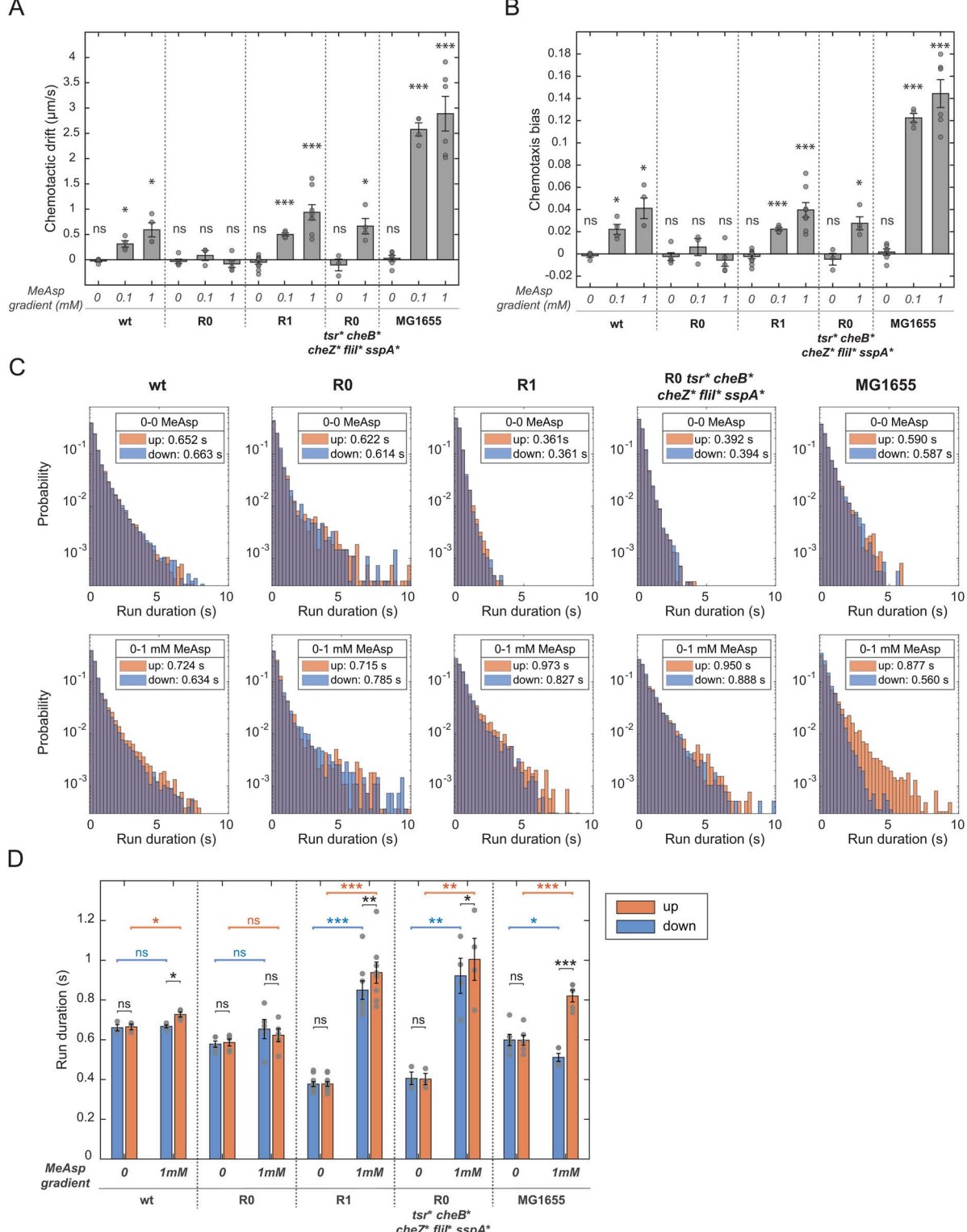

**Fig 5. Chemotactic drift and run duration of the R1 cells in a gradient of MeAsp.** (A-B) Chemotactic drift (A) and chemotaxis bias (B) of indicated strains in the MeAsp gradients quantified with cell tracking. *E. coli* MG1655 wildtype strain was used as control. Error bars represent standard error of the mean, the values for each replicate are shown with separate dots. Significance analysis was done with respect to zero using two-tailed Student

t-test. (C) Distributions of cell run durations for the indicated strains, either in the absence of a gradient (0-0 MeAsp, top panels) or in the presence of a gradient from zero to 1 mM MeAsp (0-1 mM MeAsp, bottom panels). Cell runs were measured using cell tracking and separated based on their direction, either up or down the gradient (S8A Fig, Materials and Methods). Mean run durations are shown in the insets. (D) Average run durations up and down measured in at least three replicated experiments. Error bars indicate standard error of the mean, the values for each replicate are shown with separate dots. To compare run durations up and down the gradient for the same MeAsp concentration, significance values were calculated using paired one-side Student t-test, while a non-paired Student t-test was used to compare run durations up (or down) in the absence and in the presence of the gradient. $P$ value asterisks stand for: ns, not significant; *, $P<0.05$; **, $P<0.01$; ***, $P<0.001$.

recapitulated by the $\Delta cheR$ strain carrying *tsr cheB cheZ fliI sspA* mutations, in agreement with the measurements of spreading in TBSA (Fig 3F).

Surprisingly, the drift of the R1 cells up the gradient was even slightly faster compared to the wildtype cells, which could be partly due to their higher swimming velocity. Indeed, the values of the chemotactic bias (chemotactic drift normalized by the swimming velocity) of the wildtype and R1 cells, as well as that of the cells carrying *tsr cheB cheZ fliI sspA* mutations, were very similar (Fig 5B). Thus, the chemotactic ability of the evolved R1 line is comparable to that of the non-evolved parental wildtype strain RP437. Nevertheless, these values remained below those observed for the experimentally evolved faster-swimming RP437 cells [22] or for the generally better motile MG1655 cells (Fig 5A and 5B) [48,49].

We next compared the distributions of the duration of cell runs up and down a gradient along the channel. All strains expectedly showed similar distributions in both direction in the absence of a gradient (Fig 5C, upper panels, and Fig 5D). A non-chemotactic (and nearly unresponsive) $\Delta cheR$ (R0) strain showed no significant difference between the duration of runs in either direction even in the presence of the gradient source. In contrast, runs of the wildtype cells (both RP437 and MG1655) became significantly elongated towards the source of the attractant and marginally shortened away from it (Fig 5C, lower panels, and Fig 5D). Notably, the average run duration of the wildtype cells in the gradient was similar to that in the absence of the attractant (S8C Fig). This agrees with the established chemotaxis strategy of *E. coli*, which primarily relies on the extension of cell runs up the gradient [6], above a constant average run duration that is adjusted by the adaptation system irrespectively of the current background stimulation.

The behavior of the R1 cells, and of the $\Delta cheR$ *tsr cheB cheZ fliI sspA* cells that carry the key R1 mutations, was distinctly different from that of the wildtype or $\Delta cheR$ (R0) strains. The R1 cells showed a noticeably shorter average run duration without stimulation, consistent with their increased tumbling (Fig 2A), and their runs became strongly extended both up and down the gradient (Fig 5C and 5D). Thus, in contrast to the parental $\Delta cheR$ strain, and similar to the wildtype, the R1 line is clearly responsive to the attractant. Moreover, both the R1 cells and $\Delta cheR$ *tsr cheB cheZ fliI sspA* cells showed a significantly larger elongation of runs up the gradient, which may form the basis of their chemotactic drift. But both strains are apparently unable to efficiently adjust the average cell running behavior to the background concentration of attractant (S8C Fig), consistent with the absence of CheR-mediated adaptation.

To directly test the ability of the evolved $\Delta cheR$ lines to adapt, we investigated their pathway response to attractant using the FRET assay. The wildtype cells responded to the addition and subsequent removal of different MeAsp stimuli in a characteristically dose-dependent and adaptive fashion (Fig 6A), as already discussed above (S5A Fig). No measurable FRET response could be observed for R0 cells (S9A Fig), consistent with expectedly low pathway activity in these cells. In contrast, the pathway response of R1 cells to MeAsp stimulation was well pronounced and observed in a similar concentration range as for the wildtype cells, but it showed no apparent adaptation on the time scale of minutes in our experiment (Fig 6B). Similarly non-adaptive response was observed in the R4 line (S9B Fig). Thus, the FRET assay confirmed that the evolved $\Delta cheR$ lines did not regain the ability to adapt to stimulation on the timescale relevant for the gradient sensing by swimming cells via some alternative mechanism [50].

Furthermore, the R1 cells have apparently higher pre-stimulation pathway activity compared to the wildtype cells, as suggested by their stronger response to attractant and weaker response to repellent ($NiCl_2$) (S9C, S9D and S9E Fig). This is consistent with higher tumbling bias and shorter duration of their runs in the motility buffer (Figs 2A and 5C). Moreover, the

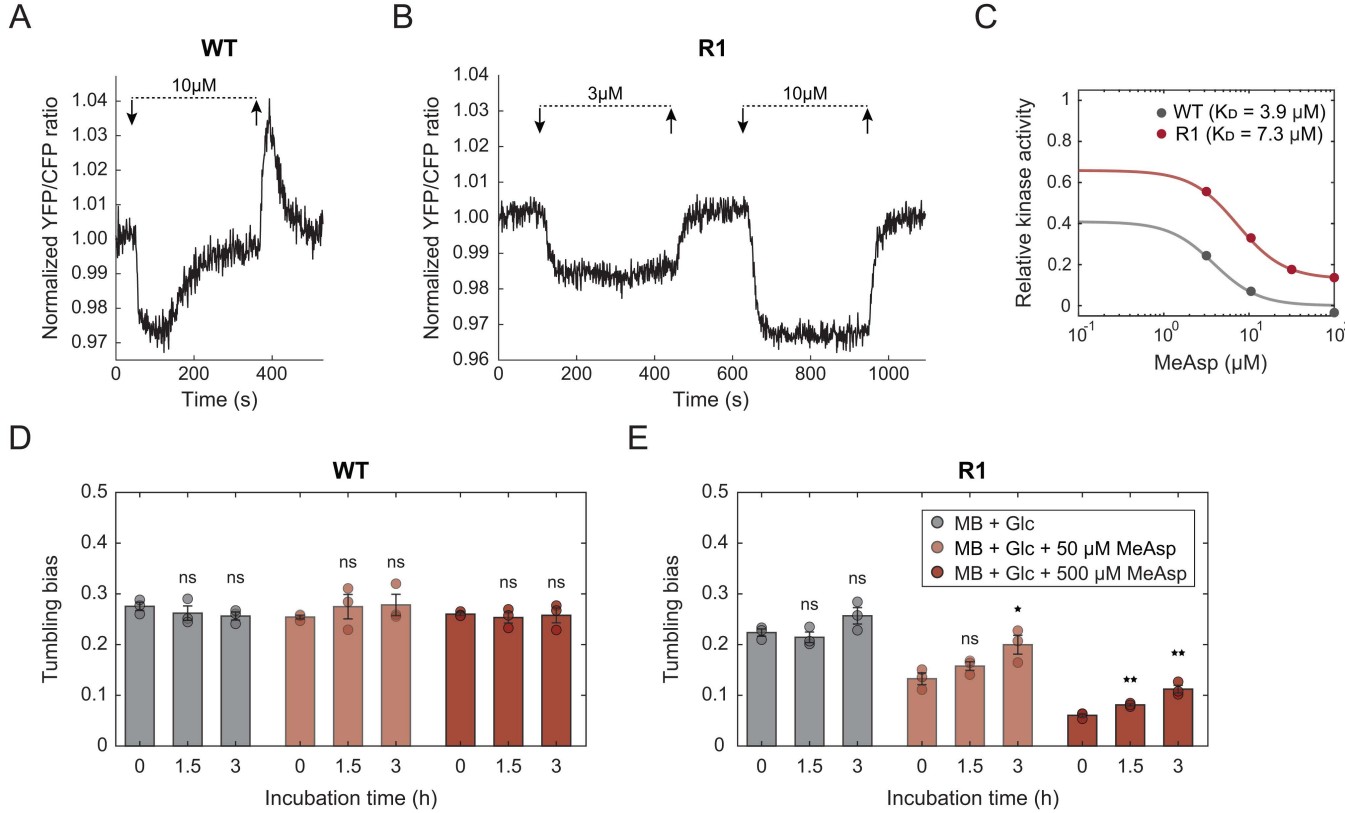

**Fig 6. Pathway response and slow adaptation of tumbling bias in the evolved R1 strain.** (A-B) YFP/CFP ratio, proportional to the chemotaxis pathway activity, for the populations of the wildtype RP437 (A) and the evolved R1 (B) cells expressing the FRET pair CheY-YFP/ CheZ-CFP. Cells were immobilized in a flow chamber under flow and exposed to the addition (down arrows) and subsequent removal (up arrows) of indicated concentrations of MeAsp. In both cases, the ratio was corrected for the baseline drift and normalized to its value at the beginning of the experiment. (C) Dependence of the relative kinase activity, calculated from initial changes in the YFP/CFP ratio (see S9 Fig and Materials and Methods), on MeAsp concentration for the wildtype and R1 cells. Experimental data was fitted by a Hill function. In case of R1, an additional parameter accounting for non-zero activity upon high MeAsp stimulations was added to the fitting function. Half-inhibitory concentrations ($K_D$) are shown in the plot. (D-E) Mean tumbling bias for the wild-type RP437 (D) or R1 (E) cells quantified with cell tracking analysis. Cells were kept in motility buffer supplemented with either only 1% glucose (Glc) or additionally with 50 μM or 500 μM MeAsp, as indicated. Motility measurements were performed in the beginning of the experiment and after 1.5 h and 3 h of incubation. Significance analysis was performed across three replicated experiments; the values for each replicate are shown with separate dots. *P* values were calculated in relation to the starting time point using two-tailed Student t-test (ns, not significant; *, $P < 0.05$; **, $P < 0.01$; ***, $P < 0.001$).

dose dependence of the R1 pathway response was moderately shifted to higher concentrations of MeAsp and, in contrast to the wildtype cells, their pathway activity was not fully inhibited by the addition of high concentrations of MeAsp (Figs 6C and S9E). These observations suggest slightly reduced sensitivity of the R1 cells to MeAsp compared to the wildtype.

The evolved Δ*cheR* cells have thus regained not only their responsiveness to chemoeffector stimulation but also the ability to differentially modulate their motility dependent on the direction in a gradient. This apparently occurs without short-term (seconds to minutes) adaptation of the pathway activity that forms the basis of the wildtype chemotaxis strategy. Nevertheless, we observed that the R1 cells are capable to at least partially recover their tumbling bias when exposed to MeAsp for prolonged periods of time (several hours), in contrast to the wildtype cells that adapt rapidly and subsequently maintain stable tumbling bias (Figs 6D and 6E, and S9F). Although such very slow adaptation is unlikely to directly contribute to the ability of the R1 cells to follow chemical gradients in liquid, it may be important for the maintenance of intermediate tumbling bias, and thus responsiveness to gradients, over a range of background attractant levels.

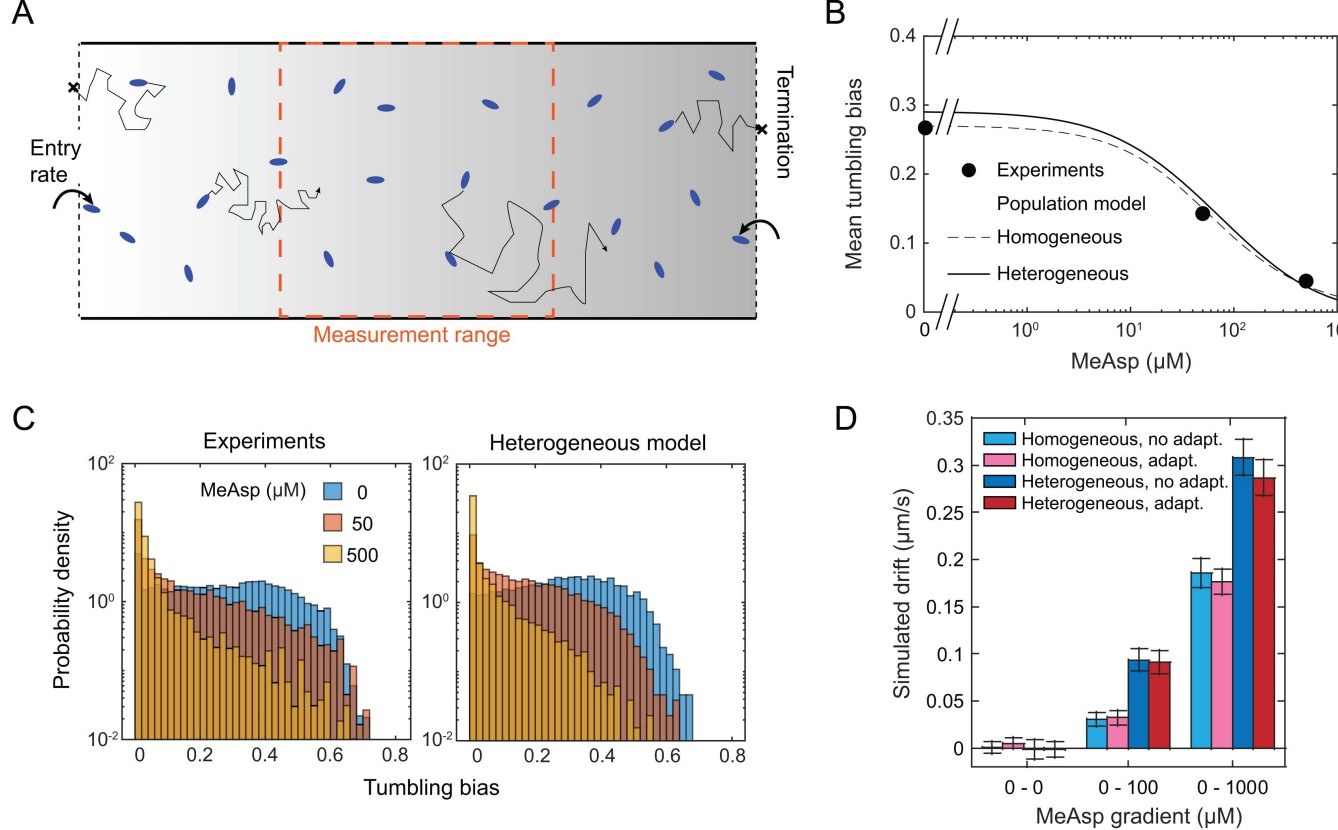

**Fig 7. Simulated non-adaptive strains with R1 tumbling properties are capable of chemotaxis-like behavior.** (A) Schematics of the simulation box featuring a chemoattractant gradient, solid walls on its sides and two open ends each connected to an infinite cell reservoir (see S1 Text for details). (B) Measured population-averaged tumbling bias of the R1 cells in the presence of various concentrations of MeAsp, fitted by the homogeneous and heterogeneous models of tumbling bias response to MeAsp. (C) Experimental distributions of tumbling bias measured in the same conditions as in (B), and the corresponding inferred distributions of tumbling bias using the heterogeneous population model. Experimental data in (B,C) are from a single representative experiment. (D) Simulated average drift of the population of cells in response to the indicated gradients of the attractant for the two models and with or without the slow adaptation, as indicated. Error bars are standard errors of the mean, with $N \simeq 1.5 \times 10^4$ trajectories per condition.

## Computational simulations can partly explain the adaptation-less taxis of the evolved ΔcheR cells

To understand whether the measured properties of the pathway and swimming responses could explain the biased tactic motion of the evolved Δ*cheR* cells, we computationally simulated the behavior of these cells in a gradient. The cells were modeled as run-and-tumbling agents which bias their tumbling probability according to the currently experienced concentration of attractant (S1 Text). To reflect our experimental setup, we simulated cell swimming in a linear gradient of MeAsp along a 2-mm long rectangular channel with open ends connected to infinite reservoirs, which absorb and randomly respawn cells at the extremities (Fig 7A). Their concentration-dependent tumbling bias was adjusted accordingly to the experimentally measured dependence on MeAsp concentration (Figs 7B and S10A). This included the observed residual tumbling bias at even high concentration of MeAsp, which is also consistent with the FRET measurements (Fig 6C). Furthermore, we either assumed a homogeneous population of cells with the same population-averaged tumbling bias or a heterogeneous population with the experimentally measured distributions of tumbling bias (Figs 7C and S10B). Notably, the experimentally measured tumbling bias distributions of both the R1 line and of our wildtype RP437 strain were broader than that of the canonical *E. coli* MG1655 strain (S10C Fig).

In our simulations, cells with either a uniform or a heterogeneous tumbling bias were able to drift on average up the gradient (Fig 7D), with higher drift speed being observed for a steeper gradient. The simulated chemotactic drift was markedly enhanced by the heterogeneity of tumbling bias within population, where individual cells are simulated to respond sharply but at different concentrations of MeAsp [51]. This enables the non-adapting population as a whole to maintain strong response over a wider range of attractant concentrations. Our simulations thus confirm that the space-dependent modulation of tumbling bias in a gradient of attractant could alone be sufficient to elicit chemotactic drift, even without short-term adaptation at the pathway level. Nevertheless, the simulated drift remained several times smaller than the one measured experimentally, even for simulations that took population heterogeneity into account. We further tested the impact of the experimentally observed gradual recovery of the tumbling bias on the timescale of hours upon prolonged attractant stimulation. However, no significant improvement of the chemotactic drift by such slow adaptation could be observed in our simulations (Fig 7D), likely because this recovery is too slow to make a direct impact on gradient sensing. The evolved Δ*cheR* cells thus appear to rely on other features that are not captured by the model, such as the dynamics of flagellar motor remodeling [52], to further enhance their ability to follow chemical gradients.

## Discussion

Experimental evolution under defined laboratory selection has been used to provide important insights into the evolvability of individual proteins, simple traits, and regulatory networks in microorganisms [53–57]. This includes the evolutionary tuning of bacterial flagellar gene expression under selection for motility and/or growth [21–24]. Here, we demonstrate the capacity of experimental evolution to reshape the signaling network that controls bacterial chemotactic behavior, resulting in the emergence of an efficient alternative strategy of chemotaxis that does not require one of the normally essential components of the pathway.

In agreement with the universal conservation of CheA, CheY, CheW and chemoreceptors (MCPs) across bacterial and archaeal phyla [58], our results suggest that the chemotaxis defects caused by the absence of these core chemotaxis signaling components could not be restored by experimental evolution. However, the deficiencies in the auxiliary signaling components, including the less conserved CheZ but also the highly conserved adaptation enzymes CheR and CheB, could be evolutionary compensated under selection for spreading in soft agar within several hundred generations.

This restoration of biased spreading has been apparently enabled by several general as well as strain-specific phenotypic and genotypic changes. One common but auxiliary change was an increase in cell swimming velocity, as had already been observed upon the evolution of enhanced chemotaxis in wildtype cells [22–24]. This increase in swimming velocity due to the upregulation of flagellar gene expression seems to be a common consequence of selection for motility. As in the previous study with the wildtype *E. coli* strain RP437 [22], it was caused by mutations in multiple genes encoding components of the flagellar export apparatus and general regulators of flagellar gene expression. Notably, although *E. coli* swimming velocity saturates at high levels of flagellar gene expression [27], several evolved strains including the R1 line exhibited expression levels higher than those that would be strictly necessary to achieve the maximal swimming velocity. Similar apparently excessive flagellar gene expression was previously observed in some strains resulting from experimental evolution of the wildtype RP437 cells for enhanced motility [22]. We hypothesize that this might represent an evolutionary "overshoot", where some of the selected mutations enhance the function beyond the optimum.

A further, more important, common phenotypic alteration was the restoration of tumbling behavior, with the tumbling bias of the evolved deletion strains becoming more similar to that of the wildtype. This phenotypic adaptation could be attributed to the importance of intermediate tumbling frequencies for the efficient spreading of motile *E. coli* in mesh-like soft agar pores [14,26]. It was achieved via mutations that compensated for the impact of the respective gene deletion on the chemotaxis pathway activity, with all evolved lines of the initially smooth-swimming Δ*cheR* strain becoming more tumbly, and most evolved lines of the initially hypertumbly Δ*cheZ* and Δ*cheB* strains becoming more smooth-swimming.

However, restoration of the tumbling bias and increase of the swimming velocity could not alone explain the evolved enhancement of spreading, as strains with similar tumbling and velocity showed distinctly different spreading efficiencies. The deletion-strain specificity of the observed compensatory mutations suggests that their effects extend beyond simple improvement of the non-directional spreading. Indeed, at least some of the evolved strains appear to have regained the capability to follow chemical gradients. This was clearly true for the several evolved ΔcheR and ΔcheZ lines that exhibited a directional bias in their spreading in chemoattractant gradients established in soft agar. Such biased spreading was particularly pronounced in the gradient of serine, where the bias of the evolved ΔcheR lines was comparable to that of the wildtype, but the spreading bias was also significant for the ΔcheR lines in the gradient of MeAsp. The difference between the two chemoeffectors may be in the steepness of the gradient experienced by cells, as serine consumption steepens the gradient at the outer edge of the spreading colony whereas MeAsp is not metabolized and forms gradient only by diffusion from the source. Additionally, our selection for the evolved spreading likely occurs in the gradient of serine, the first gradient followed by the chemotactic *E. coli* cells on TBSA plates [33]. The mutations in *tsr* observed in the ΔcheR lines might thus be selected to specifically enhance the response to serine, as indicated by the characterized phenotype of the Tsr$^{T305M}$ substitution.

Further confirming their ability to perform chemotaxis, the evolved ΔcheR cells exhibited chemotactic drift not only in soft agar but also in gradients established in liquid. The chemotactic drift of the R1 line in a stable linear gradient was comparable to or even higher than that of the non-evolved wildtype RP437, although it was apparently less efficient in accumulation towards the source of attractant. The observed drift was also slower than the one that can be achieved by the fully motile *E. coli* reference strain MG1655 [27] or by RP437 that has been evolved for enhanced motility [22]. Nevertheless, these results demonstrate a striking recovery of *E. coli* ability to perform chemotaxis even in the absence of CheR that is normally considered vital for the bacterial strategy of chemotaxis.

The tactic behavior of the evolved ΔcheR cells in chemical gradients is markedly differed from that of the wildtype cells, which indicates the emergence of a novel chemotactic strategy, or evolutionary reactivation of the pre-existing cryptic chemotaxis strategy. While the adapting wildtype cells exhibit a specific elongation of their runs up the attractant gradient and a slight shortening down the gradient, they have similar average run distributions with and without the attractant. This finding is consistent with the established paradigm of bacterial chemotaxis [6], which relies on receptor methylation to reset the steady-state pathway activity and to perform temporal comparisons of stimulation relative to this adapted level. In contrast, the chemotaxis pathway of the evolved ΔcheR cells did not regain the capacity to rapidly adapt to chemical stimulation, as demonstrated by the FRET measurements of their response. Consistent with such lack of short-term adaptability, the R1 cells extend their runs in both directions along the gradient. Nevertheless, even without adaptation they show significantly greater elongation of runs up than down the gradient, which is apparently sufficient to produce a chemotactic drift that is comparable to that of the parental wildtype strain RP437.

The observed tactic behavior could be at least partly recapitulated by our simulations, which utilized experimentally determined response parameters. We observed that, even in the absence of adaptation, the concentration-dependent decrease in the tumbling rate, which results in enhanced diffusion, can be sufficient to produce a modest bias in cell motion up a gradient of attractant. A crucial aspect that enables chemotaxis of such non-adapting cells is the combination of a sensitive yet non-saturating response across a broad range of attractant concentrations, as experimentally observed for the R1 cells. The adaptation-independent chemotactic drift could be further enhanced by the observed cell-to-cell phenotypic variability in response to attractant, which was modeled as sharp single-cell responses with a broad distribution of sensitivities to the cue [51,59]. This enables a subset of cells within the population to be sensitive to gradients across a broad range of attractant backgrounds, which might to some extent circumvent the apparent inability of individual cells to adapt to new environments.

Nevertheless, the simulations underestimated the experimentally observed chemotactic drift by at least severalfold, even when accounting for cell response heterogeneity, suggesting that the mechanistic explanation of this alternative

chemotactic behavior remains to be fully established. One potential characteristic that is not considered by our model but could contribute to chemotaxis of the non-adapting cells, is the dynamics of the stimulation-dependent flagellar motor remodeling [52]. The chemotactic response of the evolved Δ*cheR* cells might be further enhanced by the observed slow recovery of the cell tumbling bias upon stimulation. This could occur due to modulation of receptor activity over longer time scales even in the absence of CheR, by the production of half-modified receptors and their gradual deamidation by CheB. Our simulations suggest that such recovery, on the time scale of an hour, is too slow to directly contribute to the chemotactic drift. However, it may be important to counteract the response saturation and readjust the steady-state tumbling bias to the intermediate range where chemotactic response remains functional. Such partial adaptation through the balance of biosynthesis and deamidation of receptors could make an even larger contribution during spreading of growing cells in soft agar, where the rate of receptor biosynthesis is higher.

In conclusion, our findings illustrate the remarkable evolvability of the bacterial chemotaxis pathway, which allows it to circumvent the deficiencies in auxiliary but highly conserved and normally essential proteins. While relying on the same pathway reactions that are evolutionary modified to compensate for the missing component, the evolved alternative mechanism of chemotaxis is qualitatively different from the wildtype behavior. The efficiency of this alternative strategy is generally lower compared to the wildtype strategy and likely limited to a narrower range of environmental conditions, and whether it represents a genuine evolutionary innovation or a rudimentary chemotaxis strategy that was already present in the wildtype strain remains to be investigated. Nonetheless, our study underscores the capacity of short-term evolution to result in the emergence of novel behavioral strategies and encourages further work on experimental evolution of novel microbial behaviors.

## Materials and methods

### Strains and growth conditions

Derivatives of *E. coli* strain RP437 [25], used in this study as the wildtype for chemotaxis, are listed in S4 Table. Unless specified otherwise, cultures were grown at 37°C in Luria broth (LB) with shaking at 200 rpm or on LB plates containing 1.5% (w/v) agar. For spreading ring measurements, *E. coli* strains were grown at 30°C for ~16 h in Tryptone broth (TB) containing 0.27% (w/v) agar (Tryptone Broth Soft Agar - TBSA). For growth rate measurements, bacterial cultures were grown in 24-well plates with transparent bottom in a plate reader (Infinite M1000 PRO, TECAN) in TB at 30°C with shaking at 180 rpm.

### Experimental evolution of the chemotaxis mutants

Experimental evolution of *E. coli* mutant strains was conducted under selective conditions to enhance spreading on Tryptone broth soft-agar (TBSA) plates. The evolution was performed in the absence of mutagenesis and relied on the natural mutation rate of *E. coli*. The chemotaxis mutants were evolved through consecutive passages of sub-culturing in TBSA. The inoculum for the initial subculture was taken directly from glycerol stocks. For subsequent reinoculations, 2 µl cells were taken from the outer edge of the spread ring after overnight growth at 30°C on the TBSA plate and inoculated onto a fresh TBSA plate (S1A Fig). This process was repeated daily for 30 days, with additional samples taken for glycerol stocks. The resulting strains were sequenced as described below, and the variant calling analysis was done to identify the SNPs, insertion and deletion in the genome. The presence of these mutations in evolved strains at different stages of evolution was then analyzed using PCR amplification and Sanger sequencing. This also confirmed that the mutations were not present at detectable levels in the original populations of non-evolved strains.

### SNP-reintegration/genome editing

For the introduction of the point mutations, cassettes containing the *neo-ccdB* were amplified from pKD45 (S4 Table) using specific primers, which include a 50-bp homology from both sides for the desired region. This cassette was

introduced in the desired *E. coli* recipient strain containing pKD46, and strains carrying this cassette and pKD46 were then transformed with a fragment containing the desired region amplified from the evolved lines using electroporation. Transformed cells were grown in LB + 0.2% arabinose for 4–5 h at 30°C, followed by selection on M9 minimal media plates containing rhamnose as sole carbon source. Rhamnose-resistant colonies were selected and sequenced to confirm mutations.

### Re-sequencing of bacterial genomes

The genomic DNA of the evolved bacterial population were isolated using the Qiagen DNeasy Blood and Tissue kit following the manufacturer's instructions. Libraries were constructed using Nextera XT Index Kit (24 indexes, 96 samples) Illumina FC-131–1001. Sequencing was done using the Illumina HiSeq Rapid Run (2 × 150 bp paired-end run). The genomes were reassembled and mutations mapped with the DNASTAR Seqman NGen 12 software and BRESEQ pipeline, using the *E. coli* RP437 genome as the template. For identifications of mutations by Sanger sequencing, gene fragments of interest were amplified using PCR. This PCR fragment was subsequently purified and sent for Sanger Sequencing to Microsynth Seqlab GmbH.

### Gradient plate assay

Minimal A agar (0.25% agar, 10 mM $KH_2PO_4$/$K_2HPO_4$, 8 mM $(NH_4)_2SO_4$, 2 mM citrate, 1 mM $MgSO_4$, 0.1 mg mL$^{-1}$ of thiamine-HCl, 1 mM glycerol, and 40 µg mL$^{-1}$ of a mixture of threonine, methionine, leucine, and histidine) supplemented with antibiotics and inducers was used for the agar plate assay. Chemical solutions (50 mM serine or 50 mM MeAsp) were applied to the centerline of the plate and incubated at 4°C for 12–16 h to generate a chemical gradient before cell inoculation. Overnight cultures of different evolved strains were washed twice with tethering buffer and applied to the plate at a distance of 1.5 cm from the line where the chemical was inoculated. Plates were incubated at 30°C for 24–48 h. Spreading bias was quantified by measuring the distances from the inoculation point of the expanding colony to its edges up or down the chemical gradient using ImageJ software and taking their ratio.

### Microfluidics chemotaxis assay

Cells were harvested by centrifugation at 4000 rpm for 5 min and washed twice with the tethering buffer. Methyl aspartate dissolved in the tethering buffer (50 mM) was adjusted to pH 7.0. The responses of *E. coli* cells to a concentration gradient were measured using a microfluidic device illustrated in S7A Fig. *E. coli* cells were added into the well of the device (labeled as 'Cells') at the final OD$_{600}$ of 1.2–2 and equilibrated for 20 min in the observation channel. Methyl aspartate solution was added into the other well ('Compound') and allowed to diffuse into the observation channel through a porous agarose membrane, and thereby establish a concentration gradient. Fluorescence microscopy using a Nikon Ti-E microscope system with a 20X objective lens was used to detect the fluorescence intensity of cells in the observation area. The cellular response was characterized by the fluorescence intensity in the observation region (300 µm × 200 µm). Data were analyzed using ImageJ (Wayne Rasband, National Institutes of Health, USA).

### FRET assay

FRET assay was performed as described previously [41,42]. Bacterial cells were transformed with pVS88 plasmid (CheY-YFP, CheZ-CFP) for FRET experiment and were grown in TB (1% tryptone, 0.5% NaCl, pH 7.0) supplemented with 100 µg mL$^{-1}$ ampicillin and 50 µM isopropyl-β-D-thiogalactoside (IPTG) at 34°C and 275 rpm. Cells were harvested at OD$_{600}$ of 0.6 and washed twice with the tethering buffer (10 mM $KH_2PO_4$/$K_2HPO_4$, 0.1 mM EDTA, 1 µM methionine, 10 mM sodium lactate, pH 7.0). Strains were immobilized in a flow chamber and stimulated with chemoeffectors added or removed by flow. FRET measurements were performed on an upright fluorescence microscope (Zeiss Axio Imager.Z1). The YFP/CFP

ratio, calculated as the ratio between fluorescence intensities in YFP and CFP channels, was corrected for the baseline drift and normalized to the value at the beginning of an experiment. The pathway activity (kinase activity) was calculated from the YFP/CFP ratio assuming full activation (A = 1) by 300 μM NiCl$_2$ and full inactivation (A = 0) by the mix of 100 μM MeAsp and 50 μM serine, and fitted to a Hill function, $A(L) = A_0 \times (1 - L^H/(L^H + K_D^H))$, where $A(L)$ is the amplitude of the measured FRET response, $A_0$ is the pathway activity in the unstimulated case, $L$ is the attractant concentration, $H$ is the Hill coefficient and $K_D$ is the half-inhibitory concentration. In case of R1 strain, an additional parameter accounting for non-zero activity upon high methyl aspartate stimulation was introduced.

## Promoter activity assays

For promoter activity assays, *E. coli* strains, transformed with the reporter plasmid pAM109, were grown in TB supplemented with 50 μg mL$^{-1}$ kanamycin in 96-well plates at 30°C in a rotary shaker at 180 rpm. Cell fluorescence was measured using flow cytometry on BD LSR Fortessa SORP cell analyzer (BD Biosciences).

## Motility analyses

For motility analysis, *E. coli* cells were grown in 10 mL TB medium at 34°C in a rotary shaker at 275 rpm. Cells (1 mL) at mid-log phase (OD$_{600}$ = 0.6) were collected and re-suspended in 1 mL motility buffer (6.15 mM K$_2$HPO$_4$, 3.85 mM KH$_2$PO$_4$, 100 μM EDTA, 67 mM NaCl, pH 7.0, 0.01% Tween80, and 1% glucose if not otherwise stated), and then diluted in motility buffer supplemented with indicated amounts of MeAsp to an OD$_{600}$ appropriate for cell tracking (~0.02-0.03). Swimming velocity and tumbling rate were measured by cell tracking in a glass chamber using phase-contrast microscopy (Nikon TI Eclipse, 10X objective NA = 0.3, CMOS camera EoSens 4CXP, 1 px = 0.7 μm). All data were analyzed using ImageJ with custom-written particle-tracking pipeline (https://gitlab.gwdg.de/remy.colin/particletracking2) for swimming velocity, drift velocity, and tumbling rate analysis [60]. In short, detection, via intensity thresholding, and tracking, via next-frame closest proximity, of single cells in a background-subtracted movie yields a set of 2D trajectories $\left\{ \boldsymbol{r}_i(t) = (x_i(t), y_i(t)) \,\middle|\, t \in t_0 + [0, T_i - 1] \right\}_i$. Further analysis ignores trajectories shorter than 1 s and separates swimmer from non-swimmer trajectories via a criterion on diffusivity, $\frac{\left\langle \left(x_i - \langle x_i \rangle_t\right)^2 \right\rangle_t + \left\langle \left(y_i - \langle y_i \rangle_t\right)^2 \right\rangle_t}{T_i} > 0.61$ μm$^2$/s for swimmers. This threshold was chosen closely above the diffusion coefficient of non-swimmers, ~0.4 μm$^2$/s at our low densities [61], to enable the analysis of tumbly mutants. Instantaneous swimming speeds of swimmers are measured via a local fit $\boldsymbol{r}(t') = \boldsymbol{r}_0(t) + \boldsymbol{v}(t)(t' - t)$ on a 10-frame sliding window. Swimmer trajectories are split between runs and tumbles based on a criterion on the local ballisticity of the trajectory, $v_r(t) = \frac{|\boldsymbol{v}(t)|}{\Delta r(t)}$ with $\Delta r(t) = \left\langle \sqrt{\left(x_i(t'+1) - x_i(t')\right)^2 + \left(y_i(t'+1) - y_i(t')\right)^2} \right\rangle_{t'}$ calculated on the same sliding window as $\boldsymbol{v}$ (normalized threshold $v_{n,th} = 0.785$). We then use the fraction of time spent tumbling, i.e., the tumbling bias, $\frac{t_{tumble}}{t_{run} + t_{tumble}}$, where $t_X$ sums the instances of state $X$ within a trajectory (for computing distributions) or over all trajectories in a movie (for average values), to quantify tumbling. When building distributions of tumbling bias, each trajectory is weighted by its duration.

## Chemotactic drift measurements and run analysis

Chemotactic drift and bias were measured according to the protocol previously described in detail [60]. Cells were grown in TB at 34°C in a rotary shaker at 275 rpm, harvested at OD$_{600}$ = 0.5 and washed three times in the motility buffer. Two chambers of a polydimethylsiloxane (PDMS) microfluidic device were filled with cells in the motility buffer (6.15 mM K$_2$HPO$_4$, 3.85 mM KH$_2$PO$_4$, 100 μM EDTA, 67 mM NaCl, pH 7.0, 1% glucose, 0.01% Tween80) supplemented either without or with 50 μM, 100 μM, 500 μM or 1 mM of MeAsp. Trajectories of single cells were recorded in the middle of the channel between the chambers using phase-contrast microscopy (Nikon TI Eclipse, 10X objective NA 0.3, CMOS camera EoSens 4CXP) at the acquisition rate of 50 frames per second and were further analyzed with the custom ImageJ particle-tracking pipeline (https://gitlab.gwdg.de/remy.colin/particletracking2). The average chemotactic drift velocity was

calculated as $V_{chem} = \frac{\sum_i V_i T_i}{\sum_i T_i}$, where $V_i$ and $T_i$ are the mean velocity in the direction of the gradient (y) and the trajectory length for each detected cell $i$.

For the analysis of run durations, runs shorter than 5 frames (0.1 s) were removed and the direction of each run $j$ was calculated as $\alpha_j = \cos^{-1} \frac{\Delta y_j}{\sqrt{\Delta x_j^2 + \Delta y_j^2}}$, where $\Delta x_j = x_{j,end} - x_{j,start}$ and $\Delta y_j = y_{j,end} - y_{j,start}$. All runs with $|\alpha| < \pi/4$ and $3\pi/4 < |\alpha| < \pi$ were classified as *up* and *down* the gradient, respectively (S8A Fig). The distributions of run durations for both subpopulations were visualized using MATLAB.

## Simulations

Numerical simulations are described in details in S1 Text. The code is available at https://github.com/croelmiyn/ExperimentallyEvolvedStrains.

## Statistical analysis

All experiments were done in triplicate. Error bars indicate respective standard errors of the mean. Significance analysis was done in comparison to the reference strains, as indicated in figure legends. *P* values were calculated using one-tailed and two-tailed Student t-test. The values for *N*, *P*, and the specific statistical test performed for each experiment are included in the appropriate figure legend.

## Supporting information

**S1 Text. Computer simulations of the chemotactic response of evolved *cheR* strain.** (PDF)

**S1 Fig. Methodology of the evolution experiment and spreading of core chemotaxis gene deletion strains.** (A) A schematic of the evolution experiment. The cells demonstrating the fastest spreading were collected from the edge of the spreading ring and inoculated in the middle of a fresh TBSA plate. This procedure was repeated for 30 days. Created with BioRender. (B) Spreading of Δ*cheA* (A), Δ*cheY* (Y), Δ*mcp* (M) or Δ*cheW* (W) deletion strains, either before (denoted as "0") or after evolution for 30 days in soft agar. An example of one evolved line (denoted as "1") is shown for each strain. Scale bars are 2 cm. (PDF)

**S2 Fig. Size of the spreading rings for the additional evolved Δ*cheR* lines.** Spreading was measured for the indicated *cheR* lines in three independent replicates after ~16 h incubation in TBSA and normalized to the spreading of the wildtype (wt). Error bars indicate standard errors of the mean. (EPS)

**S3 Fig. Growth rates of the parental strains and the evolved Δ*cheR*, Δ*cheB* and Δ*cheZ* lines.** Optical density at 600 nm was measured every 30 mins for the indicated strains grown in TB medium at 30°C in a plate reader, and the growth rate was calculated by fitting an exponential function to the exponential phase of growth. The values were normalized by the growth rate of the wildtype strain. Significance analysis was done in relation to the corresponding non-evolved strain. *P* values were calculated using two-tailed Student t-test (ns, not significant; *, $P < 0.05$; **, $P < 0.01$; ***, $P < 0.001$). (EPS)

**S4 Fig. Order of appearance and effects of point mutations in Δ*cheR* lines.** (A-D) Spreading of R1 (A), R2 (B), R3 (C), and R4 (D) lines in TBSA over the time course of experimental evolution (in days). The first day when a respective mutation could be identified by Sanger sequencing is highlighted with the arrows. Spreading values were obtained from a single experiment and normalized to the spreading of the wildtype strain. (E) Effects of individual mutations in *cheB* and

*tsr* genes from R4 and R5 lines on spreading when introduced in the R0 strain. (F) Effects of individual R1 mutations on spreading when introduced in the wildtype background. (E, F) The diameters of spreading rings were measured in three independent replicates; error bars indicate standard errors of the mean. Significance analysis was done in comparison to R0 (E) or to wt (F). *P* values were calculated using two-tailed Student t-test (ns, not significant; *, $P<0.05$; **, $P<0.01$; ***, $P<0.001$).
(EPS)

**S5 Fig. Effects of selected *cheB* and *tsr* mutations on the pathway response.** (A-F) Pathway response to attractant stimulation, measured using FRET, for the wildtype (WT) cells (A,C,E) or the wildtype cells carrying a mutation in *cheB* (WT *cheB**; B) or in *tsr* (WT *tsr**; D,F) expressing the FRET pair CheY-YFP/ CheZ-CFP. YFP/CFP ratio, proportional to the pathway activity, was corrected for the baseline drift over time and normalized to its value at the beginning of the experiment. Indicated concentrations of MeAsp, serine and $NiCl_2$ were added (down arrows) and subsequently removed (up arrows) at the specified time points. (G) Kinase activity, calculated from the YFP/CFP ratio (see Materials and Methods) as a function of attractant (serine or MeAsp) concentration for the WT and WT *tsr** strains. The data were fitted by a Hill function. Half-inhibitory concentrations ($K_D$) and Hill coefficients ($H$) are shown.
(EPS)

**S6 Fig. Biased movement of the evolved strains towards serine.** (A-C) Indicated strains were tested for biased spreading on M9 minimal medium soft-agar (M9SA) plates with a pre-established gradient of serine (50 mM at the source). Scale bars represent 1 cm. (D) Spreading bias was quantified as the ratio between the distances from the inoculation point of the expanding colony to its edges up and down the gradient. The measurements were performed in three independent replicates; error bars indicate standard errors of the mean. Significance analysis was done in comparison to 1 (no bias). *P* values were calculated using one-tailed Student t-test (ns, not significant; *, $P<0.05$; **, $P<0.01$; ***, $P<0.001$).
(PDF)

**S7 Fig. Cell accumulation assays towards a non-metabolizable chemoattractant. (A)** A scheme of the microfluidic device used for the quantification of chemotaxis in Fig 4C. The observation area is depicted in dashed rectangle. **(B)** Exemplary images from the observation area for the indicated strains taken in the beginning and after 180 min. 50 mM MeAsp was used to form the concentration gradient.
(EPS)

**S8 Fig. Quantification of the chemotactic drift and run duration in a linear MeAsp gradient. (A)** A schematic of a polydimethylsiloxane (PDMS) microfluidic device used for the chemotactic drift measurements. Linear chemoattractant gradients in the middle channel were created by filling one chamber with cells in the motility buffer and the other one with cells in the motility buffer containing either no, or 100 μM, or 1 mM MeAsp. Trajectories of single cells were traced with particle tracking and runs were classified as "up" and "down" the attractant gradient based on their directionality. **(B)** Visualization of the gradient established in the observation channel with fluorescein. The scale bar is 500 μm. (A,B) The images are modified from [49]. **(C)** Distributions of run durations for the indicated strains, either in the absence of a gradient (MB) or in the presence of a gradient from zero to 1 mM MeAsp. All runs, regardless of their directions, were included in the distributions. The average run duration for each condition is shown in the figure legend.
(EPS)

**S9 Fig. The chemotaxis pathway response in the wildtype and the evolved Δ*cheR* strains. (A-B)** Normalized YFP/ CFP ratio for the Δ*cheR* (A) and the evolved Δ*cheR* strain R4 (B) expressing the FRET pair CheY-YFP/ CheZ-CFP. The ratio was corrected for the baseline drift and normalized to the value at the beginning of an experiment. Indicated concentrations of MeAsp were added (down arrow) and subsequently removed (up arrow). **(C, D)** FRET response to indicated

concentrations of MeAsp, combination of MeAsp, serine and $NiCl_2$ for the evolved R1 (C) or WT (D) strain. **(E)** The relative kinase activity of the pathway deduced from the YFP/CFP ratio for WT and R1 strains for different stimulations (see Materials and Methods). The response to high concentrations of MeAsp + Ser and $Ni^{2+}$ were assigned as 0 and 1 for the kinase activity of the evolved R1 strain. Shown are means (large symbols) and individual values (small symbols) from two independent experiments. **(F)** Run duration distributions for the evolved R1 cells kept in motility buffer supplemented with either only 1% glucose (Glc) or additionally with 500 µM MeAsp. The distributions from a representative experiment are plotted for the starting time point and after 3-hour incubation. The average run durations are shown in the figure legends. (EPS)

**S10 Fig. Tumbling bias of the evolved R1 strain. (A)** Population-averaged tumbling bias of the R1 strain measured by cell tracking in the presence of various homogeneous concentrations of MeAsp and for two different suspending media: our regular motility buffer containing 1% glucose (MB + Glc) and glucose-free motility buffer (MB - Glc). Experimental data are from a single representative experiment. The data are fitted by two models of tumbling bias response to MeAsp (see S1 Text). Homogeneous population: all cells respond identically. Heterogeneous population: underlying distributions of sensitivity and basal activity are assumed. **(B)** Experimental distribution of tumbling biases for the R1 strain measured in the same conditions as panel A, combining a dataset taken close to and away from the surface and shown on either logarithmic (upper panels) or linear (middle panels) scale, as well as the corresponding inferred distributions of tumbling bias, using the heterogeneous population model described in S1 Text (lower panels). **(C)** Experimental distribution of tumbling bias in the WT cells of RP437 and MG1655 backgrounds, acquired in MB + Glc and combining datasets taken close to and away from the surface, shown on either logarithmic (left panel) or linear (right panel) scale. The values of the mean tumbling bias for both strains are indicated. The increased density at the tumbling bias equals zero is due to short trajectories (1–4 s) recorded away from the surface, within which the cell passed through the field of view without tumbling. (EPS)

**S1 Table. Mutations identified in evolved Δ*cheR* lines.**
(PDF)

**S2 Table. Mutations identified in evolved Δ*cheB* lines.**
(PDF)

**S3 Table. Mutations identified in evolved Δ*cheZ* lines.**
(PDF)

**S4 Table. Strains and plasmids used in this study.**
(PDF)

**S1 Data. All genomic mutations identified in evolved strains.**
(XLSX)

**S1 File. Source data files for all figures.**
(ZIP)

## Acknowledgments

We thank Ferencz Paldy and Lars Velten for their help with preliminary experiments.

## Author contributions

**Conceptualization:** Manika Kargeti, Irina Kalita, Ron L. Dy, Remy Colin, Victor Sourjik.

**Formal analysis:** Manika Kargeti, Irina Kalita, Remy Colin.

**Funding acquisition:** Victor Sourjik.

**Investigation:** Manika Kargeti, Irina Kalita, Sarah Hoch, Maryia Ratnikava, Wenhao Xu, Bin Ni, Ron L. Dy, Remy Colin.

**Methodology:** Manika Kargeti, Irina Kalita, Remy Colin, Victor Sourjik.

**Project administration:** Victor Sourjik.

**Resources:** Irina Kalita, Remy Colin.

**Software:** Remy Colin.

**Supervision:** Victor Sourjik.

**Visualization:** Manika Kargeti, Irina Kalita, Remy Colin.

**Writing – original draft:** Manika Kargeti, Irina Kalita, Remy Colin, Victor Sourjik.

**Writing – review & editing:** Manika Kargeti, Irina Kalita, Remy Colin, Victor Sourjik.

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
