## [Decision Letter · Decision Letter 0]

PGENETICS-D-25-00061

Experimental evolution of a bacterial chemotaxis network with reduced complexity

PLOS Genetics

Dear Dr. Sourjik,

Thank you for submitting your manuscript to PLOS Genetics. After careful consideration, we feel that it has merit but does not fully meet PLOS Genetics's publication criteria as it currently stands. Therefore, we invite you to submit a revised version of the manuscript that addresses the points raised during the review process.

Please submit your revised manuscript within 60 days Apr 21 2025 11:59PM. If you will need more time than this to complete your revisions, please reply to this message or contact the journal office at plosgenetics@plos.org. Please include the following items when submitting your revised manuscript:

We look forward to receiving your revised manuscript.

Kind regards,

Jianzhi Zhang

Academic Editor

PLOS Genetics

Pablo Wappner

Section Editor

PLOS Genetics

Aimée Dudley

Editor-in-Chief

PLOS Genetics

Anne Goriely

Editor-in-Chief

PLOS Genetics

**Journal Requirements:**

3) We notice that your supplementary Figures, Tables, and information are included in the manuscript file. Please remove them and upload them with the file type 'Supporting Information'. Please ensure that each Supporting Information file has a legend listed in the manuscript after the references list.

Potential Copyright Issues:

i) Figures 1A, and S1A. Please confirm whether you drew the images / clip-art within the figure panels by hand. If you did not draw the images, please provide (a) a link to the source of the images or icons and their license / terms of use; or (b) written permission from the copyright holder to publish the images or icons under our CC BY 4.0 license. Alternatively, you may replace the images with open source alternatives. See these open source resources you may use to replace images / clip-art:

6) Thank you for stating that "Java code for numerical simulations is available at "https://github.com/croelmiyn/ExperimentallyEvolvedStrains." This link reaches a 404 error page. Please amend this to a working link.

**Reviewers' comments:**

Reviewer's Responses to Questions

**Comments to the Authors:**

**Please note that the reviews are uploaded as attachments.**

Reviewer #1: The authors carried out a very thorough investigation of the ability to “evolve” restored chemotaxis function for strains deleted for one of the six Che genes (CheA, CheB, CheR, CheW, CheY, and CheZ) or all four (or five if Aer was included) chemoreceptors of E. coli. As far as I can tell, no artificial mutagenesis was conducted.

All four replicates of 30 rounds of selection for increased spreading in tryptone soft agar of the �cheR, �cheB, and �cheZ evolved lines that showed increased spreading. In contrast, none of the four replicates with �cheA, �cheW, �cheY, �cheRB, or �receptor strains evolved lines with increased spreading. The result with the �cheA, �cheW, �cheY, and �receptor is expected, as they do not make CheY-P, which is essential for CW flagellar rotation of the wild-type motor. Thus, they are locked in CCW rotation, or smooth swimming, and cannot spread in soft agar because they get trapped in cul de sacs in the agar mesh. The result with the �cheRB strain is a bit more difficult to understand on face value, because they should be able to tumble.

cheR strains do not spread because they also turn their flagella almost exclusively CCW and swim smoothly. �cheB and �cheZ strains do not spread because they turn their flagella predominately CW and thus never form a coherent bundle of left-handed helical flagellar filaments that can make the cells swim in a translational fashion (run). However, they have the gene products required to make CheY-P, and the control of the production of CheY-P by the chemotaxis system can be biased to make levels of CheY-P that allow the run-tumble behavior that is required for spreading in soft agar. Indeed, in all twelve of the evolved lines that showed spreading, four for each of �cheR, �cheB, and �cheZ, the cells were seen to run and tumble.

The most interesting result was with the lines derived from �cheR. At least the R1 line not only spread, it also showed a sharp ring at the edge of the spreading colony, which is characteristic of the ability to carry out the chemotaxis to L-serine mediated by the Tsr chemoreceptor. This indicates that cells in this line not only can run and tumble to spread in soft agar, they can also carry out chemotaxis. This was confirmed by showing that these cells can migrate up static concentration gradients generated in a flow device.

I have these observations.

1) The knowledge that cells that are missing essential components of the Che system can spread in soft agar if they can run and tumble is at least 40 years old, as the authors acknowledge. What is new here is that the authors have identified the mutations, typically multiple ones with additive effects, that restore the ability to run and tumble. They are precisely the mutations that one might have predicted. For the �cheR lines they are mutations that would be expected to increase CheY-P production by increasing CheA activity or increasing the ability of chemoreceptors to stimulate CheA activity. For the �cheB and �cheZ lines, they are mutations that should decrease the activity of CheA or the stimulation of CheA activity and thus the overproduction of CheY-P. In each case, running and tumbling is restored. The evolved lines also often contained mutations that increase flagellar gene expression, so that the cells swim faster.

2) The most interesting case is the derived lines from the �cheR mutant that can actually carry out chemotaxis to move in gradients of chemoattractants. It has been thought that the combined activities of CheR and CheB are required for cells to make the temporal comparisons in attractant concentration that allow them to move in chemical gradients. It would seem that CheR is not absolutely necessary for this. The authors provide a speculative mechanism and develop a computer program to test their model. Their program can get cells to move in chemical gradients, but not as well as the �cheR evolved lines can. To be flippant, they have shown that evolution can outperform computer programmers. I would like to see them develop a bit more completely and clearly their model for how the �cheR evolved line manages to carry out real chemotaxis. I suspect this involves the fact that receptors are synthesized with several of their methylatable glutamate residues as glutamines that are deamidated by CheB, which could provide a sort of one-shot adaptation for a receptor. The �cheR evolved strains all retain some CheB activity, and the �cheRB strain never evolved lines that can swarm. The residual activity of CheB in a �cheR strain can presumably still be regulated by phosphorylation of CheB by CheA, which suggests that enough adaptation mediated by changes in CheB activity may be possible to accomplish some degree of chemotaxis when cells are able to run and tumble.

3) My biggest criticism is that the authors seem to take credit for replicating millions of years of evolution that fine-tuned a chemotaxis system by evolving a different mechanism for chemotaxis through directed laboratory evolution over a month or two. I find that unjustified. What they have actually done is to remove a piece of a highly evolved and very adaptable chemotaxis signaling network and then restored a lesser level of function by multiple mutations in chemotaxis and flagellar genes that compensate for the loss of the removed component. The evolved systems still do not function as well as the wild-type system even under controlled laboratory conditions. I predict they would perform even less well than the wild-type system in the real world, where chemical gradients are transient in time and irregular in space. I think the authors should openly explain this and not claim to have generated sped-up evolution in a petri plate.

4) An analogy may reinforce my point from the previous paragraph. A person losing a leg will initially be virtually immobile. However, even without an artificial limb, that person will learn how to get around reasonably well by crawling, hopping, using a crutch, or finding some other means of non-bipedal locomotion. (Yes, now they could get a very sophisticated artificial leg.) I think it is inappropriate to say that they have evolved or developed a new means of locomotion, as all of those latent skills were present before the leg was lost. They just had not been required to be developed, but they could be refined and to some extent repurposed.

5) My conclusion from the points raised in the two preceding paragraphs is that the title and emphasis of the paper should be changed, and its claimed impact be put into perspective. It is an inelegant alternative title, and something better wording can no doubt but found, but more accurate would be “Loss of Adaptive Methylation by CheR Can Be Partially Compensated by Multiple Mutations Targeting Other Components of the Chemotaxis Signaling Pathway.”

6) It is also misleading to say that the system has lower complexity. The compensation for the loss of CheR, CheZ, or CheB, came through the accumulation of multiple mutations in other components of the chemotaxis and flagellar system that had to be finely balanced and likely made the system far less robust and flexible. The newly evolved chemotaxis system is almost certainly inferior to the one it replaced. Simply removing one component does not make the system less complex if multiple changes in the remaining components are required to have it function even somewhat effectively. Helicobacter pylori can get by with a simpler system because it lives in a rather simple environment.

Minor editorial comments.

Line 565. The “circumvents” should be “circumvent.”

Lines 879-880. Rewrite “…where the corresponding individual chemotaxis gene was deleted, either before…or after evolution for 30 days.” The gene was not deleted before or after 30 days of evolution. The evolved strain with the deleted gene was evaluated before and after evolution for 30 days.

Reviewer #2: In this study, the authors investigated how E. coli adapts to the loss of key genes in the CheYZ pathway, which is critical for chemotaxis. They generated knockout strains, conducted experimental evolution, and analysed evolved mutants through sequencing, RNA-seq, and detailed characterization of swimming behaviour. The results showed that the evolved mutants partially restored certain aspects of chemotaxis, but not all, concluding that experimental evolution can lead to novel behaviour strategies.

I found the study to be very interesting, but I believe the authors may have overreached with their conclusions relative to the data presented and need to better align the strength of their interpretations with the evidence.

My comments:

Abstract - I am unclear about the authors' statement that E. coli is assumed to possess the 'minimal number of components for chemotaxis.' Does this simply mean that E. coli can perform chemotaxis and therefore has the necessary components, or is there an assumption that it has the minimal possible number of components? If the latter, this assumption should be justified, as it is not obvious why we should assume there is no redundancy in the system.

Additionally, the authors claim that the complexity of the chemotaxis signalling pathway can be reduced through experimental evolution. However, it is not clear whether they have demonstrated a reduction in complexity or simply a reduction in the number of components. These are not synonymous, as complexity encompasses more than just the number of parts—it includes interactions, regulation, and system wide outputs. The authors should clarify their interpretation of 'complexity' and explain how their claims align with the evidence provided."

Line 152 can the authors please rewrite to make this a bit clearer, for example perhaps explain the phenotype of the mutants first. Then in separate sentence explain the phenotype of the evolved strains, including some comment on the variation between replicates and also some statistical reporting on the difference between the outcomes for different mutants.

Line 156 what is a core component. Is it a component that, after it has been deleted, evolution cannot restore the function? or is there some other definition.

Line 185 It is not clear to me what conclusions are being referred to here. I presume the result was consistent with what was observed in this study, but not exactly the same, so please explain what the result was that was observed, and why is this worth mentioning.

Line 191 “correlated well” is a bit vague - can the authors please supply R values and P values. Also, it would be interesting to know if this is something that has been observed before. In other words is it typical that tumbling correlates with spreading.

L198 the author's conclusion is too strong here. This is interesting, but circumstantial evidence - you do not consistently see elevated fliC expression, and you have not done the molecular genetic follow up experiments to really nail down the result. That is fine, but I think the authors should say that these data suggest or support the hypothesis rather than using words like “confirm” or “likely”.

L223 I am not sure what this sentence means. “These mutations may promote the active state of Tsr or Tar, contributing to the suppression of low receptor activity that is characteristic for the cheR strain.” It makes sense for me until it gets to the part “that is characteristic for the cheR strain.” Can the authors please write to make the meaning more clear? – perhaps unpacking this in to more that one sentence will help.

L223 to L230 – this section has quite a few ideas that are not quite fully explained. For example this sentence “Mutations in tsr may have been preferentially selected over those in tar either because the first gradient followed by chemotactic cells on TBSA plates is formed by the Tsr ligand serine (28), or because cells have higher levels of Tsr compared to Tar under our growth conditions.” is difficult to understand. Please break this up into multiple sentences and make sure there is enough information for the uninitiated to understand.

L224 – The authors write that “These mutations may promote the active state of Tsr or Tar,”. This seems very unlikely to be true for the TAR gene. There is a very important difference between the TSR and the TAR mutations - basically all of the TAR mutations cause a major loss of function, either a frameshift or an early stop. This tells us that what has been selected in these strains is a loss of function of this protein. This is pretty convincing evidence since the TSR mutants are all amino acid substitutions, in contrast. I think the authors should discuss what this means and modify their hypothesis about the impact of mutations in these two genes. Or alternatively explain in the response what I'm missing.

L251 – does this refer to all deltaCheZ lines? Or all lines in the study? Also did the authors start the study with clones? This is typical in evolution experiments, to reduce the chance that parallel evolution gets mistaken for standing genetic variation. Were there any other instances where the same exact mutation was seen in multiple different lines? Even if they were at a low frequency, the fact remains that they they spread to a high frequency during the course of the experiments suggesting that they were selected. What is the point that the authors are making here, in other words what are they suggesting when they point out that these particular mutants may have been already in the population before the study?

Methods: I would like the authors to provide more information about the provenance of their mutants - were they made for the study or were they made some time ago? In other words, has there been time to accumulate mutations in the lab, since they were made. Also, did the authors sequence the mutants constructed mutants, or just compare the evolved strains to the ancestor? And also what was the process of setting up the experiment was each replicate line a separate clone or were replicate lines founded from the same clone (or at least, single colony). These details will be important for people doing evolution experiments and also for understanding the results of the sequencing.

There is a method described for making point mutations - is this method for making the knockouts?

Figure 1 - the first panel depicting the different components of signal transduction in motility are too small and it's not clear which components correspond to which gene. For panel B the series of photos purporting to show different levels of chemo tactic capacity are a bit unclear. Panels C DE are organised in an odd way - shouldn't you just show the deletion mutant 1st and then show the results of the evolution experiment? I found the arbitrary use of color a little confusing - why some panels blue and some red, and then others in gray?

Reviewer #3: The authors deleted the chemotaxis genes from the E. coli genome one by one and then examine how chemotaxis might be recovered via evolution. They uncover several compensatory mechanisms that reestablish the ability to climb chemical gradients, revealing what aspects of the pathway are essential and which one are not. Notably they show that chemotaxis can take place without adaptation mechanisms by only controlling the effective diffusion coefficient of cells (mean tumble bias) as a function of the absolute concentration of ligand, decreasing it as ligand concentration goes up.

The powerful E. coli chemotaxis system is ideal for this study and experiments are carried out carefully. Some of the analysis needs some work, especially the measurements of tumble bias, which in wild type look very different from what is measured in other studies. Overall this is an excellent study that should be of interest to the broad readership of PLoS Genetics.

Main comments:

Figure 1C and elsewhere: How is the spreading ring size normalized? It appears that the authors divide the size of the evolved mutants’ rings by that of the WT ring. However, is the WT strain also evolved, or is only the non-evolved WT used for normalization? The authors previously demonstrated that the spreading rate of RP437 increases significantly after ~6 rounds of evolution (Ni et al., 2017).

Figure 2A: Line 189: unlike for CheR and CheZ, for CheB it is not really the case that most evolved lines reduced tumbling to WT levels. Can the authors comment on this? Why is CheB different? Is it due to the extra regulation of CheB via phosphorylation?

Figure 2D: The promoter activity of fliA, the upstream regulator of fliC, decreases with increasing growth rate (Honda et al., 2022). Could the observed difference in fliA promoter activity be explained by a slower growth rate in the evolved strains? Did the authors measure growth rate in these strains, and if so, does it correlate with spreading rate?

Figures 2C and 2D: Strain R1 exhibits ~2.5× higher fliC expression than strain R2, yet both strains have nearly identical run speeds. The authors previously showed a strong correlation between fliC expression and swim speed (Lisevich et al.). Could this indicate that R2 has reached the fliC expression threshold above which the run speed plateaus? If so, do the authors believe the increased fliC expression in the evolved strains provides any functional advantage?

Lines 229-230: The authors state that E. coli has higher levels of Tsr than Tar under their growth conditions. However, their 2010 study showed that RP437 has more Tsr than Tar molecules only at ODs < 0.1. Given that cell density is presumably high on agar plates, do the authors have Western blot data to support this claim?

Line 251 and Table S3: The evolved ΔcheZ strains all carry tar mutations, but none have tsr mutations. If tar translation is completely disrupted, cells would predominantly express Tsr receptors, leading to lower kinase activity. Could the enhanced spreading rate in these strains be attributed to increased serine sensitivity (due to Tsr dominating the kinase response) combined with reduced kinase activity (due to the absence of Tar), leading to lower CheY-P levels compensating for the lack of CheZ?

Lines 251-256: Unlike for the DeltaCheB, the authors do not provide a rational for the mutation they observe in the deltaCheZ lines. Why deletion of CheZ that results in high CheY-P would lead to mutation that interrupt Tar translation? How does that lead to reduction of kinase activity?

Likewise is the mutation in the P1 domain of CheA compensating for the reduced dephosphorylation of CheY by reducing phosphorylation rate?

Line 307: the authors conclude that the CheB activity is reduced. Is there direct evidence of that?

Figures 4 and S4: Many evolved mutants form a ring when chemotaxing toward serine (Figure S4), yet none form a ring toward MeAsp (Figure 4), whereas the WT forms a ring in both cases. Why is that?

Figure 6 and S8: the distribution of tumble bias is extremely wide in Figs 6C, S8B and even for the wt RP437 (Fig s8C): the distribution is almost uniform up to 0.6. Several previous studies have shown that the distribution of tumble bias in that same strain is typically much narrower with a peak around 0.2-0.3 with almost no cells with tumble bias above 0.5. Is this an artefact of the tracking or analysis algorithm? Did the authors count also cells that barely move, or count multiple tumbles within one effective tumble? It is difficult to assess as not much information is provided about the quantification of tumble bias from individual tracks.

Modeling. does the modeling predict a maximal stimulus where the population can no longer chemotax? Presumably, once the kinase activity is maximally suppressed, there are no longer position-dependent changes in tumbling rate.

Figure S7E. the dynamic range of the kinase response appears larger than the wild-type, which is in line with the interpretation that chemotaxis is made possible by extending the concentration range over which tumble bias varies. Do the authors know if the receptor or kinase mutations affect the cooperativity of the chemoreceptor arrays? How do these mutations affect WT receptor arrays?

Minor comments:

Line 120: drop the word nutrients? E.g. aspartate and serine are not really nutrients

Figure 1: Scale bar in panel B is missing. C-E: What does 2, 3, and 4 mean in e.g. B2 B3 B4. Are these are replicates of te evolutionary experiment starting with the same ancestor?

Line 191: claim that tumbling correlates with spreading in TBSA, but no correlation is calculated in the text or the figure caption

Line 193: Claim ‘nearly all’ evolved strains have increased swim speed. Should clarify that this is w.r.t. the deletant strains, because the significance in Figure 2C is calculated w.r.t. the wild-type.

Figure 3C: the text makes reference to receiver domains, and methylesterase domains. Could these be labeled like the receptor domains in 3AB?

Figure 4B: I’m a bit confused on how this bias is calculated. The authors say it’s the ratio between ‘spreading’ up and down the gradient, but how is this spreading calculated? Is this just the furthest point reached in either direction, or does this take into account the overall shape of the cell density?

**Have all data underlying the figures and results presented in the manuscript been provided?**

Reviewer #1: Yes

Reviewer #2: Yes

Reviewer #3: Yes

PLOS authors have the option to publish the peer review history of their article (what does this mean? ). If published, this will include your full peer review and any attached files.

**Do you want your identity to be public for this peer review?** For information about this choice, including consent withdrawal, please see our Privacy Policy .

Reviewer #1: No

Reviewer #2: No

Reviewer #3: No

**Figure resubmission:**
---

## [Decision Letter · Decision Letter 1]

PGENETICS-D-25-00061R1

Experimental evolution partially restores functionality of bacterial chemotaxis network with reduced number of components

PLOS Genetics

Dear Dr. Sourjik,

Thank you for submitting your manuscript to PLOS Genetics. After careful consideration, we feel that it has merit but does not fully meet PLOS Genetics's publication criteria as it currently stands. Therefore, we invite you to submit a revised version of the manuscript that addresses the points raised during the review process.

Please submit your revised manuscript within 30 days Jul 11 2025 11:59PM. If you will need more time than this to complete your revisions, please reply to this message or contact the journal office at plosgenetics@plos.org. Please include the following items when submitting your revised manuscript:

We look forward to receiving your revised manuscript.

Kind regards,

Jianzhi Zhang

Academic Editor

PLOS Genetics

Pablo Wappner

Section Editor

PLOS Genetics

Aimée Dudley

Editor-in-Chief

PLOS Genetics

Anne Goriely

Editor-in-Chief

PLOS Genetics

**Additional Editor Comments:**

Two of the original reviewers re-reviewed your manuscript. They are largely satisfied with your revision, with only a few minor issues remaining. Please constructively address them.  

**Reviewers' comments:**

Reviewer's Responses to Questions

**Comments to the Authors:**

Reviewer #2: The authors have addressed all of my concerns, and their rewrites make the paper much clearer. I have no future comments.

Reviewer #3: The authors have done a great job at addressing my questions in detail and have updated the figures accordingly. I have a few remaining minor questions:

The authors now compare the tumble bias of MG1655 and RP437, saying that the experimentally measured tumbling bias of the wildtype RP437 strain was “markedly broader than that of the canonical E. coli MG1655 strain (Figure S10C)”. However, in Figure S10c the distributions for RP437 and MG1655 look quite similar. The use of a log scale for the vertical axis might be misleading when talking about the width of these distributions. Could the authors also plot them with the vertical axis on a linear scale? In general, in might be more informative to plot the tumble bias distributions using a linear scale for the vertical axis throughout the paper.

There seems to be a high density near zero tumble bias in the RP437 distribution, which is surprising for the wild type RP437. Could this be an artifact? A minimal track duration of 1 s is very short. A typical value needed for accurate determination of tumble bias is 10 s. Also are some of the cells circling near the surface, which suppresses tumbles? Finally, if tracks have different lengths, the tumble bias of each track should be weighted by the duration of the track when generating the distribution to avoid over-counting low tumble bias cells.

**Have all data underlying the figures and results presented in the manuscript been provided?**

Reviewer #2: Yes

Reviewer #3: Yes

PLOS authors have the option to publish the peer review history of their article (what does this mean? ). If published, this will include your full peer review and any attached files.

**Do you want your identity to be public for this peer review?** For information about this choice, including consent withdrawal, please see our Privacy Policy .

Reviewer #2: No

Reviewer #3: No

**Figure resubmission:**
---

## [Editor Report · Decision Letter 2]

Dear Dr Sourjik,

We are pleased to inform you that your manuscript entitled "Experimental evolution partially restores functionality of bacterial chemotaxis network with reduced number of components" has been editorially accepted for publication in PLOS Genetics. Congratulations!

Yours sincerely,

Jianzhi Zhang

Academic Editor

PLOS Genetics

Pablo Wappner

Section Editor

PLOS Genetics

Aimée Dudley

Editor-in-Chief

PLOS Genetics

Anne Goriely

Editor-in-Chief

PLOS Genetics

Comments from the reviewers (if applicable):

**Data Deposition**

http://datadryad.org/submit?journalID=pgenetics&manu=PGENETICS-D-25-00061R2

**Press Queries**

---

## [Editor Report · Acceptance letter]

PGENETICS-D-25-00061R2

Experimental evolution partially restores functionality of bacterial chemotaxis network with reduced number of components

Dear Dr Sourjik,

We are pleased to inform you that your manuscript entitled "Experimental evolution partially restores functionality of bacterial chemotaxis network with reduced number of components" has been formally accepted for publication in PLOS Genetics! Your manuscript is now with our production department and you will be notified of the publication date in due course.

With kind regards,

Judit Kozma

PLOS Genetics

On behalf of:
